# ReBind: Enhancing Ground-state Molecular Conformation Prediction via Force-Based Graph Rewiring

**Taewon Kim**[1,2*]**, Hyunjin Seo**[1,2*]**, SungSoo Ahn**[1]**, Eunho Yang**[1,3†]
Korea Advanced Institute of Science and Technology (KAIST)[1], Polymerize[2], AITRICS[3]
`{maxkim139, bella72, sungsoo.ahn, eunhoy}@kaist.ac.kr`

## Abstract

Predicting the ground-state 3D molecular conformations from 2D molecular graphs is critical in computational chemistry due to its profound impact on molecular properties. Deep learning (DL) approaches have recently emerged as promising alternatives to computationally-heavy classical methods such as density functional theory (DFT). However, we discover that existing DL methods inadequately model inter-atomic forces, particularly for non-bonded atomic pairs, due to their naive usage of bonds and pairwise distances. Consequently, significant prediction errors occur for atoms with low degree (*i.e.*, low coordination numbers) whose conformations are primarily influenced by non-bonded interactions. To address this, we propose ReBind, a novel framework that rewires molecular graphs by adding edges based on the Lennard-Jones potential to capture non-bonded interactions for low-degree atoms. Experimental results demonstrate that ReBind significantly outperforms state-of-the-art methods across various molecular sizes, achieving up to a 20% reduction in prediction error. The code is available in:
https://github.com/holymollyhao/ReBIND

## 1 Introduction

The ground-state conformation of a molecule represents the lowest energy state on the potential energy surface, where the inter-atomic forces are balanced at equilibrium. This 3D conformation of the molecule plays a crucial role in determining the molecule's physical, chemical, and biological properties. As a result, it is utilized in various applications such as molecular property prediction (Satorras et al., 2021; Schütt et al., 2021; Liu et al., 2022; Thölke & De Fabritiis, 2022; Zhou et al., 2023a; Zaidi et al., 2023; Ni et al., 2024), drug discovery (Luo et al., 2021; Ganea et al., 2021; Jing et al., 2022; Xu et al., 2022; Zhou et al., 2023b; Tang et al., 2024), and protein-ligand interactions (Pei et al., 2024; Wang et al., 2024).

Recently, deep learning approaches (Hu* et al., 2020; Xu et al., 2021c; Brody et al., 2022; Rampasek et al., 2022; Xu et al., 2024) have emerged as promising alternatives to reduce the computational costs of ab initio calculations such as density functional theory (DFT; Kohn & Sham, 1965; Parr et al., 1979). These approaches focus on predicting 3D molecular conformations by leveraging graph neural networks (GNNs) with 2D molecular graphs as input. Central to most of these models is the assumption that a 2D molecular graph, where atoms are represented as nodes and covalent bonds as edges, can effectively capture atomic interactions. Building on this assumption, they employ iterative message-passing updates to predict molecular conformations, which are parallel to the iterative force-based updates used for conformer optimization. Specifically, GNN-based methods rely on covalent bonds (Hu* et al., 2020; Brody et al., 2022; Rampasek et al., 2022) and pairwise distances (Xu et al., 2024) during the message-passing phase to represent inter-atomic forces, encouraging bonded and proximate atomic pairs to be close in the representational space.

In this work, we challenge the assumption of existing approaches that bonds and inter-atomic distances are sufficient to capture the complex behaviors of atomic interactions. In reality, forces are

---

[*]Equal Contribution.
[†]Corresponding Author.

influenced by factors *beyond* bonds and monotonic distance properties, especially for non-bonded atomic pairs that are typically dominated by van der Waals potentials (Lu & Chen, 2020; Lii & Allinger, 1989). Consequently, significant prediction errors are observed for existing GNNs on atoms with low degree, *i.e.*, low coordination numbers, whose conformations can be more sensitive to non-bonded interactions, as illustrated in Figure 1 in Section 3. Therefore, a new model to accurately account for the atomic interactions is warranted.

In response to this challenge, we present REBIND, a novel framework that rewires molecular graphs by adding edges between non-bonded atomic pairs exhibiting high inter-atomic forces, often modeled with the Lennard-Jones (LJ) potential (Jones, 1924). To this end, we propose a force-aware self-guidance graph transformer, whereupon the initial conformation predicted from the encoder, a LJ potential-based adjacency matrix is constructed, then utilized in the decoder to predict the final conformation.

To be specific, utilizing the encoder's prediction of inter-atomic distances from the initial conformation, REBIND calculates the absolute forces acting between all non-bonded atomic pairs through the derivative of the LJ potential. Subsequently, REBIND augments the graph by adding edges to non-bonded pairs with the largest computed forces. Our edge augmentation is performed in a degree-compensating manner, ensuring that atoms with fewer connections receive additional edges to enhance the modeling of its non-bonded interactions. Furthermore, to differentiate the nature of these interactions, we distinguish the augmented edges into distinct adjacency matrices each specifically modeling repulsive and attractive forces. Incorporating these force-based adjacency matrices, the decoder refines the initial conformation prediction by generating residual adjustments, leading to a more precise molecular geometry.

The versatility of our proposed framework is demonstrated through benchmarks on both small-scale datasets, *i.e.*, QM9 (Ramakrishnan et al., 2014) and Molecule3D (Xu et al., 2021c), and a large-scale GEOM-DRUGS (Axelrod & Gomez-Bombarelli, 2022) dataset. We also show that our idea of force-based rewiring brings universal improvements to GNNs even outside the proposed architecture, i.e., GINE (Hu* et al., 2020), GATv2 (Brody et al., 2022), and GraphGPS (Rampasek et al., 2022).

Our contributions are summarized as follows:

- We reveal that current approaches, which solely rely on bonds and predicted pairwise distances, are insufficient for accurately modeling inter-atomic forces, especially resulting in significant errors for nodes with low-degree atoms.

- We introduce REBIND, a novel graph rewiring framework that adds edges between non-bonded atomic pairs with high inter-atomic forces guided by the Lennard-Jones potential. The number of augmented edges for each atom is determined in a degree-compensating fashion, improving the modeling of non-bonded interactions for low-degree atoms.

- Extensive evaluation on diverse molecular sizes demonstrates the effectiveness of REBIND in enhancing ground-state molecular conformation prediction. Notably, our framework achieves improvements of up to 20% on the QM9 dataset.

## 2 PRELIMINARIES

**Problem definition.** We focus on predicting the 3D ground-state molecular conformation from its corresponding 2D molecular graph $\mathcal{G} = (\mathcal{V}, \mathcal{E})$, where $\mathcal{V}$ denotes the set of $N = |\mathcal{V}|$ atoms (nodes) and $\mathcal{E}$ represents the set of $M = |\mathcal{E}|$ bonds (undirected edges) between atomic pairs. Nodes are characterized by a feature matrix $\boldsymbol{X} = [\boldsymbol{x}_1, \boldsymbol{x}_2, ..., \boldsymbol{x}_N]^\mathsf{T} \in \mathbb{R}^{N \times d}$, where each feature encodes atomic properties such as atom types and chirality. Edges are described by a binary adjacency matrix $\boldsymbol{A} \in \mathbb{R}^{N \times N}$, where $\boldsymbol{A}[i, j] = 1$ if a bond exists between atoms $i$ and $j$, and $\boldsymbol{A}[i, j] = 0$ otherwise. Additionally, an edge feature matrix $\boldsymbol{E} = [\boldsymbol{e}_1, \boldsymbol{e}_2, ..., \boldsymbol{e}_M]^\mathsf{T} \in \mathbb{R}^{M \times f}$ may be utilized to represent bond-specific attributes, including bond types. For each node $i$ in a single graph, its degree is notated as $\deg_i = \sum_{j \in \mathcal{V}} \mathbb{1}[\boldsymbol{A}[i, j] = 1]$ and relative degree is defined as $\deg_i^{\mathrm{rel}} = \deg_i / \max_{n \in [1, N]} \deg_n$. The ground-state molecular conformation is represented as $\boldsymbol{C} = [\boldsymbol{C}_1, \boldsymbol{C}_2, ..., \boldsymbol{C}_N]^\mathsf{T} \in \mathbb{R}^{N \times 3}$, where each $\boldsymbol{C}_i \in \mathbb{R}^3$ corresponds to the 3D coordinate of atom $i$. The atomic pairwise distance is denoted as $\boldsymbol{D} \in \mathbb{R}^{N \times N}$, where $\boldsymbol{D}_{ij} = \|\boldsymbol{C}_i - \boldsymbol{C}_j\|_2$.

**Multi-head self-attention.** At the core of a transformer (Vaswani, 2017) is a self-attention mechanism, which allows each instance in the input data to attend to every other instances, thereby enabling the model to capture the instance-wise relationships. Formally, given input feature matrix $\boldsymbol{X}$, the self-attention computes query ($\boldsymbol{Q}$), key ($\boldsymbol{K}$), and value ($\boldsymbol{V}$) matrices through linear transformations:

$$\boldsymbol{Q} = \boldsymbol{X}\boldsymbol{W}^Q, \quad \boldsymbol{K} = \boldsymbol{X}\boldsymbol{W}^K, \quad \boldsymbol{V} = \boldsymbol{X}\boldsymbol{W}^V, \tag{1}$$

where $\boldsymbol{W}^Q, \boldsymbol{W}^K, \boldsymbol{W}^V \in \mathbb{R}^{d \times d_k}$ are learnable weight matrices. The attention scores at are then computed as follows:

$$\text{Attention}(\boldsymbol{Q}, \boldsymbol{K}, \boldsymbol{V}) = \sigma_{\text{sm}}\left(\frac{\boldsymbol{Q}\boldsymbol{K}^\mathsf{T}}{\sqrt{d_k}}\right)\boldsymbol{V}, \tag{2}$$

where $\sigma_{\text{sm}}$ is a softmax function and the scaling factor $\sqrt{d_k}$ stabilizes the gradients during training.

To enhance the model's ability to capture diverse patterns, multi-head attention employs multiple attention heads in parallel. Each head independently performs self-attention, and their outputs are concatenated and linearly transformed as:

$$\text{MultiHead}(\boldsymbol{Q}, \boldsymbol{K}, \boldsymbol{V}) = \text{Concat}(\boldsymbol{O}_1, \ldots, \boldsymbol{O}_H)\boldsymbol{W}^O, \tag{3}$$

where each $\boldsymbol{O}_h = \text{Attention}(\boldsymbol{Q}_h, \boldsymbol{K}_h, \boldsymbol{V}_h)$ and $\boldsymbol{W}^O \in \mathbb{R}^{Hd_k \times d_{\text{model}}}$ is a learnable weight matrix.

**Inter-atomic interaction modeling in prior works.** Traditional message-passing-based architectures (Hu* et al., 2020; Brody et al., 2022; Rampasek et al., 2022) leverage covalent bonds to model interactions between connected atomic pairs. For a given node $i$ at the $l$-th layer, the node representation $\boldsymbol{h}_i$ is updated via message aggregation from a set of its bonded neighboring nodes $\mathcal{N}_i$, as formulated below:

$$\boldsymbol{h}_i^{(l+1)} = \psi\left(\boldsymbol{h}_i^{(l)}, \ \phi\left(\{\boldsymbol{h}_j^{(l)}, \boldsymbol{g}_{ij}^{(l)} \mid j \in \mathcal{N}_i\}\right)\right), \tag{4}$$

where $\phi$ is an aggregation function that combines the representations of neighboring nodes, and $\psi$ is an update function that integrates this aggregated information with the node's current hidden state. $\boldsymbol{g}_{ij}^{(l)}$ denotes the hidden representation of the edge feature corresponding to connected node pairs.

Recently, a new encoder-decoder based graph transformer (Xu et al., 2024) was proposed to capture inter-atomic forces by utilizing both bonds and atomic pairwise distances. In this framework, the pairwise distance matrix $\widehat{\boldsymbol{D}} \in \mathbb{R}^{N \times N}$ is computed from the initial conformation predicted by the encoder. For each $h$-th head in the decoder, the adjacency matrix $\boldsymbol{A}$ representing bond existence and a row-subtracted Euclidean distance matrix $\boldsymbol{D}^{\text{row-sub}}$, defined as $\boldsymbol{D}_{ij}^{\text{row-sub}} = \max_{n \in [1,N]} \widehat{\boldsymbol{D}}_{in} - \widehat{\boldsymbol{D}}_{ij}$, are integrated as residuals in the multi-head attention score $\widehat{\boldsymbol{S}}_h$, formulated as follows:

$$\begin{aligned}
\widehat{\boldsymbol{S}}_h &= \boldsymbol{S}_h + \boldsymbol{S}_h \odot (\beta_h^A \times \boldsymbol{A}) + \boldsymbol{S}_h \odot (\beta_h^D \times \boldsymbol{D}^{\text{row-sub}}), \\
\boldsymbol{S}_h &= \boldsymbol{Q}_h \boldsymbol{K}_h^\mathsf{T} = (\boldsymbol{Z}\boldsymbol{W}_h^Q)(\boldsymbol{Z}\boldsymbol{W}_h^K)^\mathsf{T},
\end{aligned} \tag{5}$$

where $\boldsymbol{S}_h \in \mathbb{R}^{N \times N}$ is the original self-attention score computed by the outer product between key and query matrices $\boldsymbol{Q}_h, \boldsymbol{K}_h \in \mathbb{R}^{N \times d_k}$, and $\beta_h^A, \beta_h^D \in \mathbb{R}$ are learnable parameters that determine the influence of bonds and distance factors, respectively.

## 3 LIMITATION OF INTER-ATOMIC FORCE MODELING IN PRIOR STUDIES

In this section, we highlight the shortfalls of current approaches in using covalent bonds (Hu* et al., 2020; Brody et al., 2022; Rampasek et al., 2022) or pairwise distances (Xu et al., 2024) to model inter-atomic forces, followed by experimental results supporting our claim.

**Limitations of prior works for non-bonded interactions.** In ground-state molecular conformation prediction, accurately modeling inter-atomic forces is essential, as the net force on each atom defined by the gradient of molecular energy $E$, approaches zero at equilibrium (Shi et al., 2021a). This equilibrium condition primarily determines the spatial arrangement of atoms. Meanwhile, a classic model of the energy $E$ divides the term into bonded and non-bonded interactions (Leach,

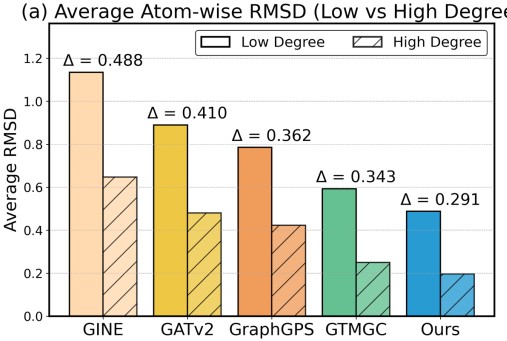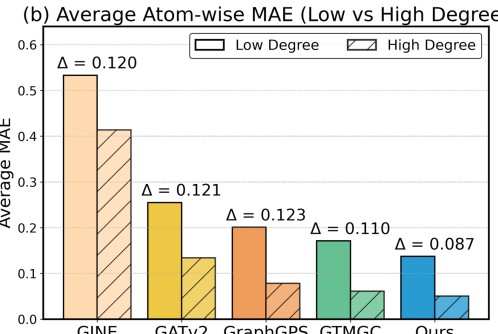

Figure 1: Atom-wise error analysis with respect to the relative atom degree on varing architectures in the QM9 dataset. (a) represents the average atom-wise RMSD while (b) shows the average atom-wise MAE for each bin of low-degree and high-degree atoms. $\Delta$ denotes the gap between the errors of the low-degree and high-degree groups.

2001; Luo et al., 2021), where bonded interactions can be further divided into bond stretching, angle bending, and torsion as shown in Equation 6.

$$E = E_{\text{bonded}} + E_{\text{non-bonded}} = E_{\text{bond}} + E_{\text{angle}} + E_{\text{torsion}} + E_{\text{non-bonded}} \quad (6)$$

We find that previous works inadequately account for non-bonded interactions. As shown in Section 2, these methods incorporate the bond adjacency matrix $\boldsymbol{A}^{\text{bond}}$ to capture the bonded interactions $E_{\text{bonded}}$ (Hu* et al., 2020; Brody et al., 2022; Rampasek et al., 2022), or the distance-based proximity matrix $\boldsymbol{D}^{\text{row-sub}}$ (row-subtracted Euclidean distance matrix) to approximate non-bonded interactions $E_{\text{non-bonded}}$ (Xu et al., 2024). However, even the distance-based proximity matrix $\boldsymbol{D}^{\text{row-sub}}$ of GT-MGC, which explicitly aims to model non-bonded interactions, oversimplifies non-bonded forces by assuming a monotonic relationship between attention weights and $\boldsymbol{D}^{\text{row-sub}}$. In reality, non-bonded interactions are governed by complex, non-linear functions of distance with atom type-specific coefficients (Leach, 2001).

Consequently, this abstraction leads to performance degradation in atoms dominated by non-bonded potential $E_{\text{non-bonded}}$, and furthermore, $E$. We believe that atoms with fewer covalent bonds (*i.e.*, low-degree atoms) correspond to such cases, due to being associated with fewer bonded potentials and hence being more sensitive to inaccuracies in the modeling of non-bonded interactions. We empirically verify this assertion in the subsequent paragraph.

**Experimental verification.** To validate our assertion, we conduct an analysis on how prediction errors vary with respect to their node degree. Specifically, we calculated an atom-wise Root Mean Square Deviation (RMSD) and Mean Absolute Error (MAE) of conformation predictions on the QM9 dataset. The calculation of atom-wise RMSD and MAE for each atom $i$ are as follows:

$$\text{RMSD}(i) = \|\widehat{\boldsymbol{C}}_i - \boldsymbol{C}_i\|_2, \quad \text{MAE}(i) = \frac{1}{N-1} \sum_{j \in \mathcal{V} \setminus \{i\}} |\widehat{\boldsymbol{D}}_{ij} - \boldsymbol{D}_{ij}|_1$$

Subsequently, we categorize nodes into low-degree ($\deg_i^{\text{rel}} \in (0, 0.3]$) and high-degree ($\deg_i^{\text{rel}} \in [0.7, 1]$) groups based on their relative degree $\deg_i^{\text{rel}} = \deg_i^{(m)} / \max_{n \in [1,N]} \deg_n^{(m)}$, then visualize the average errors for each group in Figure 1. We evaluate upon GINE, GATv2 and GraphGPS which utilize covalent bond adjacency $\boldsymbol{A}^{\text{bond}}$ and GTMGC which additionally incorporates inter-atomic distance $\boldsymbol{D}^{\text{row-sub}}$. As illustrated, the low-degree atoms exhibit significantly higher errors compared to high-degree atoms, with deviations up to 0.488 for $\text{RMSD}(i)$ and 0.123 for $\text{MAE}(i)$. This demonstrates that previous works fail to sufficiently model inter-atomic forces between non-bonded atomic pairs, hindering the effective modeling of $E_{\text{non-bonded}}$.

## 4 REBIND: ENHANCING GROUND-STATE MOLECULAR CONFORMATION VIA FORCE-BASED GRAPH REWIRING

Given the limitation of previous studies, we introduce REBIND, a novel graph rewiring framework that selectively adds edges in a force-aware manner, prioritizing atoms with low-degree. In Sec-

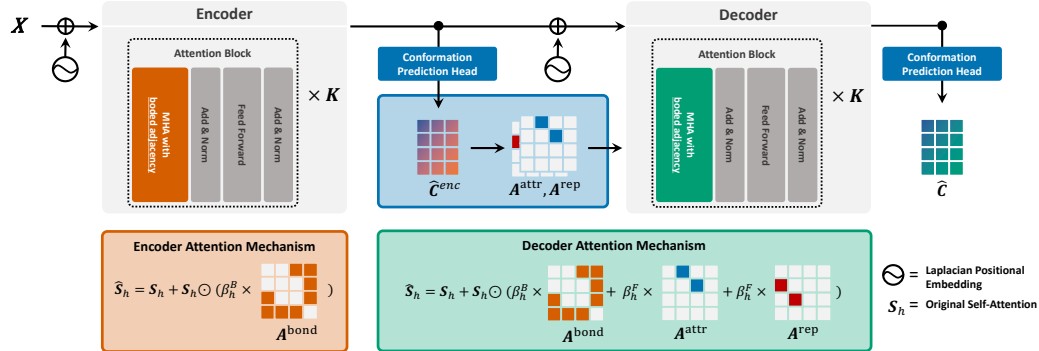

Figure 2: Overview of the REBIND framework.

tion 4.1, we provide an overview of the REBIND architecture. Section 4.2 details the force-aware graph rewiring component, and Section 4.3 describes the integration of the augmented edges into the multi-head self-attention. Finally, the learning objective of REBIND and comparison with existing studies are detailed in Section 4.4.

## 4.1 OVERVIEW

The overall architecture of REBIND is illustrated in Figure 2. Our framework receives a 2D molecular graph as input, characterized by its bonded adjacency matrix $A^{\text{bond}}$ and node feature or embeddings augmented with Laplacian positional encoding $L$. The framework outputs a predicted molecular conformation, denoted as $\widehat{C}$. We employ a standard encoder-decoder architecture with multi-head self-attention, following the approaches presented in (Vaswani, 2017; Cai & Lam, 2020; Xu et al., 2024). In this setup, the encoder processes the input graph to generate hidden representations $H^{\text{enc}}$. A task-specific prediction head then produces an initial conformation prediction $\widehat{C}^{\text{enc}}$ from $H^{\text{enc}}$. From this intermediate prediction, we derive 1) an inter-atomic distance matrix $\widehat{D}^{\text{enc}}$ and 2) force-aware adjacency matrices $\mathcal{A}^{\text{force}} = \{A^{\text{attr}}, A^{\text{rep}}\}$, where $A^{\text{attr}}$ and $A^{\text{rep}}$ denote adjacency matrices identifying attractive and repulsive atomic pairs. The hidden representation $H^{\text{enc}}$, along with $A^{\text{bond}}$ and $\mathcal{A}^{\text{force}}$, are subsequently fed into the decoder to generate the refined, residual molecular representation.

## 4.2 FORCE-AWARE GRAPH REWIRING

Here, we outline the construction of force-aware adjacency matrix $\mathcal{A}^{\text{force}}$, which connects non-bonded atomic pairs exerting significant forces to one another, mitigating the shortfalls of prior works identified in Section 3.

**Force modeling with Lennard-Jones potential.** To capture the interactions between non-bonded atomic pairs, we leverage the Lennard-Jones (LJ) potential (Jones, 1924), a well-established model for non-bonded interactions such as van der Waals forces. The LJ potential is defined as:

$$V(r) = 4\varepsilon \left[ \left( \frac{\sigma}{r} \right)^{12} - \left( \frac{\sigma}{r} \right)^{6} \right], \tag{7}$$

where $r$ is the distance between two atoms, $\varepsilon$ is the depth of the potential well (representing the strength of the interaction), and $\sigma$ is the finite distance at which the inter-atomic potential is zero. The first term $\left( \frac{\sigma}{r} \right)^{12}$ accounts for the repulsion when atoms are too close, while the second term $\left( \frac{\sigma}{r} \right)^{6}$ accounts for the attraction at moderate distances. By using the derivative of this potential, we compute the inter-atomic forces acting upon non-bonded atom pairs, which is critical for modeling the equilibrium conformation of the molecule. This force model provides a more realistic representation of how spatial positions are influenced, especially for low-degree atoms, where non-covalent interactions dominate.

The values of $\sigma$ and $\varepsilon$ are assigned based on the specific pair of atom types being modeled (*e.g.*, carbon, hydrogen, etc.). In line with conventional force-field modeling approaches used in com-

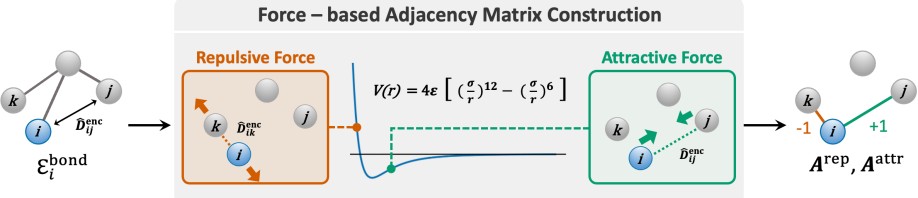

Figure 3: An illustration of edge augmentation in REBIND using the LJ potential. Non-bonded atomic pairs with the largest force magnitudes are added as edges to the molecular graph in a degree-compensating manner. The augmented edges are treated according to the nature of the forces, distinguishing between attractive and repulsive interactions.

putational chemistry, we adopt the predefined parameter values from the Universal Force Field (UFF) (Rappé et al., 1992). For interactions between different atom types, the parameters are determined using the Lorentz-Berthelot mixing rules. Specifically, the interaction parameters $\sigma_{ij}$ and $\varepsilon_{ij}$ for atom pair $i$ and $j$ are defined as follows:

$$\sigma_{ij} = \frac{\sigma_i + \sigma_j}{2}, \; \varepsilon_{ij} = \sqrt{\varepsilon_i \cdot \varepsilon_j} \tag{8}$$

Thus, the full inter-atomic force between atom $i$ and $j$ can be written as:

$$F(r) = -\frac{dV(r)}{dr} = 24\varepsilon_{ij} \left[ 2 \left( \frac{\sigma_{ij}}{r} \right)^{12} - \left( \frac{\sigma_{ij}}{r} \right)^6 \right] \frac{1}{r} \tag{9}$$

Here, the sign of $F(r)$ denotes the directionality of the force, where positive values correspond to repulsion, and negative values indicate attraction.

**Construction of force adjacency matrix.**   Given the derived force function from the LJ potential, we construct a set of force adjacency matricesdj $\mathcal{A}^{\text{force}}$ which introduces additional edges to atoms that significantly impact its spatial conformation, while prioritizing those of low-degree. Utilizing the predicted distance matrix obtained from the encoder $\widehat{\boldsymbol{D}}^{\text{enc}}$, where $\widehat{\boldsymbol{D}}^{\text{enc}}_{ij} = \|\boldsymbol{C}_i - \boldsymbol{C}_j\|_2 = \sqrt{(\boldsymbol{C}_i - \boldsymbol{C}_j)^\top (\boldsymbol{C}_i - \boldsymbol{C}_j)}$, we compute the inter-atomic force $\boldsymbol{F}^{\text{LJ}}$ for non-bonded atomic pairs using the Equation 9. We then augment the graph by connecting each node $i$ to the top $\mathrm{K}_i = \max_{n \in [1,N]} \deg_n^m - \deg_i{}^m$ non-bonded atoms that exert the largest forces on $i$ as follows:

$$\mathcal{E}_i^{\text{force}} = \left\{ j \mid j \in \mathrm{Top}_{\mathrm{K}_i} \left( \{ j \mid |\boldsymbol{F}^{\text{LJ}}[i,j]|\} \right), \; \boldsymbol{e}_{ij} \notin \mathcal{E}^{\text{bond}} \right\} \tag{10}$$

Here, for each atom $i$, we select non-bonded neighbors $j$ that exert the largest forces on $i$, adding up to the maximum possible degree $\max_{n \in [1,N]} \deg_n^m - \deg_j^m$ without thresholding or introducing any hyperparameters. To differentiate the nature of these forces, we decompose $\mathcal{E}_i^{\text{force}}$ into separate adjacency matrices $\boldsymbol{A}^{\text{attr}}$ and $\boldsymbol{A}^{\text{rep}}$ which capture attraction and repulsion, respectively. With $\mathbb{1}$ as the indicator function, we formalize $\boldsymbol{A}^{\text{attr}}$ and $\boldsymbol{A}^{\text{rep}}$ as follows:

$$\boldsymbol{A}_{ij}^{\text{attr}} = \mathbb{1}\left[ (i,j) \in \mathcal{E}_i^{\text{force}} \wedge \boldsymbol{F}^{\text{LJ}}[i,j] < 0 \right], \boldsymbol{A}_{ij}^{\text{rep}} = -\mathbb{1}\left[ (i,j) \in \mathcal{E}_i^{\text{force}} \wedge \boldsymbol{F}^{\text{LJ}}[i,j] \geq 0 \right]$$

Note that the edge weights of $\boldsymbol{A}^{attr}$ are positive ones (*i.e.* $+1$) while weights of $\boldsymbol{A}^{rep}$ are negative ones (*i.e.* $-1$), ensuring distinctiveness between the directionality of forces. The complete process is outlined in Figure 3.

### 4.3   INTEGRATION WITH MULTI-HEAD SELF-ATTENTION

Leveraging the augmented edges, REBIND enhance the multi-head self-attention by incorporating both bond-based and force-based adjacency matrices. Specifically, the augmented force-based adjacency matrices $\mathcal{A}^{\text{force}} = \{\boldsymbol{A}^{\text{attr}}, \boldsymbol{A}^{\text{rep}}\}$ and original bond-based adjacency matrix $\boldsymbol{A}^{\text{bond}}$ are incorporated as residual components in the attention scores for each head $h$. This integration is formalized as follows:

$$\widehat{\boldsymbol{S}}_h = \boldsymbol{S}_h + \boldsymbol{S}_h \odot (\beta_h^B \times \boldsymbol{A}^{\text{bond}}) + \boldsymbol{S}_h \odot (\beta_h^{Attr} \times \boldsymbol{A}^{\text{attr}}) + \boldsymbol{S}_h \odot (\beta_h^{Rep} \times \boldsymbol{A}^{\text{rep}}), \tag{11}$$

where $\boldsymbol{S}_h = \boldsymbol{Q}_h \boldsymbol{K}_h^\top = (\boldsymbol{Z}\boldsymbol{W}_h^Q)(\boldsymbol{Z}\boldsymbol{W}_h^K)^\top$ represents the global self-attention score. Here, $\boldsymbol{Z} = \boldsymbol{H}^{\text{enc}} + \boldsymbol{L} \in \mathbb{R}^{N \times d}$ is the input to the decoder, combining the encoder's hidden representations $\boldsymbol{H}^{\text{enc}}$

with Laplacian positional encoding $\boldsymbol{L}$. $\beta_h^B, \beta_h^{Attr}, \beta_h^{Rep}$ are learnable parameters that modulate the influence of 1) bonds, 2) attraction and 3)repulsion adjacency matrices, respectively. Subsequently, the output of each attention head, $\boldsymbol{O}_h$, is computed by applying a softmax normalization to the scaled attention scores and then multiplying by the linear projection of the decoder input $\boldsymbol{V}_h \in \mathbb{R}^{N \times d_k}$:

$$\boldsymbol{O}_h = \sigma_{\mathrm{sm}}\left(\frac{\widehat{\boldsymbol{S}}_h}{\sqrt{d_k}}\right)\boldsymbol{V}_h, \quad \boldsymbol{V}_h = \boldsymbol{Z}\boldsymbol{W}_h^V, \tag{12}$$

where $\sigma_{\mathrm{sm}}$ is a softmax function applied to the attention score scaled by $\sqrt{d_k}$. The final conformation $\widehat{\boldsymbol{C}}$ is obtained by refining the initial encoder representation $\boldsymbol{H}^{\mathrm{enc}}$ with the residual decoder representation $\boldsymbol{H}^{\mathrm{dec}}$, which is achieved via channel-wise attention:

$$\widehat{\boldsymbol{C}} = \mathrm{FFN}(\boldsymbol{Y}) \in \mathbb{R}^{N \times 3}, \quad \boldsymbol{Y} = \sum_{i=1}^{2} \boldsymbol{\alpha}_{:,:,i} \odot \boldsymbol{H}_{:,:,i} \in \mathbb{R}^{N \times d_c},$$
$$\boldsymbol{\alpha} = \sigma_{\mathrm{sm}}\left(\frac{\boldsymbol{H}^{\mathrm{enc}}\boldsymbol{W}_y \,||\, \boldsymbol{H}^{\mathrm{dec}}\boldsymbol{W}_y}{\sqrt{d_c}}\right) \in \mathbb{R}^{N \times d_c \times 2}, \quad \boldsymbol{H} = \boldsymbol{H}^{\mathrm{enc}} \,||\, \boldsymbol{H}^{\mathrm{dec}} \in \mathbb{R}^{N \times d_{\mathrm{model}} \times 2}, \tag{13}$$

where $\mathrm{FFN}(\cdot)$ is a task-specific head layer, $\boldsymbol{W}_y \in \mathbb{R}^{\mathrm{model} \times d_c}$ is a linear transformation applied to the hidden representations, and $||$ denotes concatenation along the last dimension. The channel-wise attention weights $\boldsymbol{\alpha}$ determine the contribution of the encoder and decoder representations in producing the final conformation.

Throughout this attention mechanism, REBIND effectively models inter-atomic forces by learning from both bonded and non-bonded atomic pairs. By incorporating the structural supervision from $\boldsymbol{A}^{\mathrm{bond}}$ and the force-aware supervision from $\mathcal{A}^{\mathrm{force}}$ into the global attention scores $\boldsymbol{S}$, REBIND facilitates comprehensive force modeling, enhancing the accuracy of conformation predictions.

### 4.4 OBJECTIVE FUNCTION

To ensure that the predicted molecular conformation remains invariant to rotation and translation, we adopt a loss function based on the difference between the predicted and ground-truth pairwise Euclidean atomic distances, denoted as $\widehat{\boldsymbol{D}}$ and $\boldsymbol{D}$, respectively, which is also utilized in prior works (Xu et al., 2021c; 2024). Additionally, to achieve precise force modeling within the decoder, we apply the same loss objective to the initial conformation predictions $\widehat{\boldsymbol{D}}^{\mathrm{enc}}$ and $\boldsymbol{D}$ from the encoder. The overall loss objective $\mathcal{L}$ is formulated as follows:

$$\mathcal{L} = \frac{1}{N^2} \sum_i^N \sum_j^N |\widehat{\boldsymbol{D}}_{ij}^{\mathrm{enc}} - \boldsymbol{D}_{ij}| + \frac{1}{N^2} \sum_i^N \sum_j^N |\widehat{\boldsymbol{D}}_{ij} - \boldsymbol{D}_{ij}| \tag{14}$$

**Comparison with existing graph transformers.** Several existing works integrate inter-atomic relationships as residuals within multi-head self-attention mechanisms. For instance, the Geometric Transformer (Choukroun & Wolf, 2021) utilizes the pairwise Euclidean distances $\boldsymbol{D}$, computed from ground-truth atomic coordinates, to perform molecular property prediction. Similarly, MAT (Maziarka et al., 2020) incorporates both the pairwise distances $\boldsymbol{D}$ and the bond adjacency matrix $\boldsymbol{A}^{\mathrm{bond}}$ for the same task. In contrast, GTMGC (Xu et al., 2024) employs predicted inter-atomic distances $\boldsymbol{D}^{\mathrm{row-sub}}$ to forecast ground-state molecular conformations.

While our work also leverages pairwise atomic relationships as residuals in the computation of attention scores, REBIND distinguishes itself by enabling fine-grained force modeling. This is achieved through the joint utilization of both bonded interactions and non-bonded interactions characterized by the largest forces. Consequently, our framework attains more precise ground-state molecular geometries, which is substantiated in the subsequent section.

## 5 EXPERIMENTS

### 5.1 EXPERIMENTAL SETUP

**Datasets.** We evaluated REBIND on well-established benchmark datasets, including QM9 (Ramakrishnan et al., 2014), Molecule3D (Xu et al., 2021c), and GEOM-DRUGS (Axelrod & Gomez-Bombarelli, 2022). QM9 consists of small organic molecules and is widely utilized for quantum

Table 1: Conformer prediction performance and percentage reduction (%) of REBIND compared to the best baseline performance on the QM9 and Molecule3D datasets.

| Splits | | Validation | | | | Test | | | |
|---|---|---|---|---|---|---|---|---|---|
| Datasets | Methods | D-MAE↓ | D-RMSE↓ | C-RMSD↓ | E-RMSD↓ | D-MAE↓ | D-RMSE↓ | C-RMSD↓ | E-RMSD↓ |
| | RDKit-DG | 0.328 | 0.570 | 0.502 | 1.044 | 0.330 | 0.573 | 0.504 | 1.266 |
| | RDKit-ETKDG | 0.324 | 0.574 | 0.458 | 1.048 | 0.325 | 0.574 | 0.460 | 1.120 |
| | GINE | 0.605 | 0.950 | 0.865 | 1.703 | 0.606 | 0.946 | 0.867 | 1.696 |
| | GATv2 | 0.382 | 0.695 | 0.711 | 1.371 | 0.382 | 0.690 | 0.712 | 1.358 |
| QM9 | GraphGPS (RW) | 0.328 | 0.629 | 0.628 | 1.196 | 0.327 | 0.624 | 0.628 | 1.193 |
| | GraphGPS (LP) | 0.283 | 0.500 | 0.544 | 1.049 | 0.283 | 0.499 | 0.546 | 1.064 |
| | GTMGC | 0.280 | 0.470 | 0.415 | 0.792 | 0.281 | 0.471 | 0.414 | 0.800 |
| | REBIND | **0.252** | **0.442** | **0.320** | **0.601** | **0.254** | **0.446** | **0.321** | **0.610** |
| | Reduction ↑ | 10.00 | 5.96 | 22.89 | 24.12 | 9.61 | 5.31 | 22.46 | 23.75 |
| | RDKit-DG | 0.581 | 0.930 | 1.043 | 1.864 | 0.582 | 0.932 | 1.044 | 1.872 |
| | RDKit-ETKDG | 0.575 | 0.942 | 0.981 | 1.700 | 0.576 | 0.943 | 0.983 | 1.710 |
| | DeeperGCN-DAGNN | 0.509 | 0.849 | N/A | N/A | 0.571 | 0.961 | N/A | N/A |
| Molecule3D | GINE | 0.591 | 1.016 | 1.103 | 2.227 | 0.592 | 1.019 | 1.104 | 2.230 |
| (random) | GATv2 | 0.564 | 0.985 | 1.072 | 2.163 | 0.565 | 0.989 | 1.073 | 2.168 |
| | GraphGPS (RW) | 0.512 | 0.900 | 1.006 | 2.089 | 0.513 | 0.903 | 1.008 | 2.094 |
| | GraphGPS (LP) | 0.440 | 0.730 | 0.854 | 1.687 | 0.441 | 0.732 | 0.854 | 1.689 |
| | GTMGC | 0.429 | 0.713 | 0.708 | 1.347 | 0.430 | 0.715 | 0.709 | 1.350 |
| | REBIND | **0.418** | **0.706** | **0.698** | **1.314** | **0.419** | **0.708** | **0.699** | **1.317** |
| | Reduction ↑ | 2.56 | 0.98 | 1.41 | 2.45 | 2.56 | 0.98 | 1.41 | 2.44 |
| | RDKit-DG | 0.542 | 0.872 | 0.993 | 1.751 | 0.524 | 0.857 | 0.970 | 1.780 |
| | RDKit-ETKDG | 0.531 | 0.874 | 0.916 | 1.565 | 0.511 | 0.858 | 0.892 | 1.595 |
| | DeeperGCN-DAGNN | 0.617 | 0.930 | N/A | N/A | 0.763 | 1.176 | N/A | N/A |
| Molecule3D | GINE | 0.889 | 1.517 | 1.398 | 3.007 | 1.388 | 2.200 | 1.928 | 4.142 |
| (scaffold) | GATv2 | 0.791 | 1.402 | 1.264 | 2.675 | 1.248 | 2.082 | 1.768 | 3.832 |
| | GraphGPS (RW) | 0.503 | 0.853 | 0.978 | 2.047 | 0.595 | 1.024 | 1.065 | 2.203 |
| | GraphGPS (LP) | 0.417 | 0.690 | 0.827 | 1.639 | 0.411 | 0.690 | 0.827 | 1.606 |
| | GTMGC | 0.406 | 0.670 | 0.682 | 1.318 | 0.397 | 0.671 | 0.692 | 1.275 |
| | REBIND | **0.391** | **0.661** | **0.640** | **1.174** | **0.386** | **0.663** | **0.667** | **1.182** |
| | Reduction ↑ | 3.69 | 4.20 | 6.16 | 10.93 | 2.77 | 1.19 | 3.61 | 7.29 |

chemistry applications. Molecule3D is a large-scale dataset of molecular structures, for which we employed two distinct splitting strategies: random split and scaffold split. The scaffold split groups molecules based on their core substructures, enabling a more realistic evaluation. Additionally, GEOM-DRUGS comprises large-size molecules relevant to drug discovery, providing a challenging benchmark for assessing the scalability of our framework on complex molecular structures. Since the original dataset includes multiple stable conformations, we choose the most stable conformation with respect to the Boltzmann energy for each molecule. Detailed descriptions of each dataset are provided in Appendix C.

**Metrics.** Following previous work (Xu et al., 2024), we evaluate the performance of our model using Mean Absolute Error (MAE), Root Mean Squared Error (RMSE), and Root Mean Square Deviation (RMSD). Additionally, we introduce a new metric, Energy-weighted RMSD (E-RMSD), which accounts for the chemical feasibility of predicted conformations. By denoting $\boldsymbol{G}_i$ and $\boldsymbol{G}$ as the *aligned* predicted and ground-state conformation via the Kabsch algorithm (Kabsch (1978)), E-RMSD is calculated as:

$$\text{E-RMSD}(\widehat{\boldsymbol{G}}, \boldsymbol{G}) = \frac{p}{\widehat{p}} \sqrt{\sum_{i \in \mathcal{V}} w_i \|\widehat{\boldsymbol{G}}_i - \boldsymbol{G}_i\|_2} \tag{15}$$

Here, $\frac{p}{\widehat{p}}$ denotes the Boltzmann factor denoted as $\exp\left(\frac{\widehat{E}-E}{kT}\right)$, while atom-wise normalized force $w_i$ is defined as $w_i = \frac{F_i}{\sum_{j \in \mathcal{V}} F_j}$. $E$ and $\widehat{E}$ denotes the total energy of ground-truth and predicted conformations, and $F_i$ denotes the force calculated via the force-field defined from the predicted conformation, both computed using the Merck Molecular Force Field (Halgren, 1996). The constants $k$ and $T$ are each the Boltzmann constant and thermodynamic temperature, respectively. This approach penalizes 1) errors of molecules that are energetically unstable via $\frac{p}{\widehat{p}}$, and 2) errors of atoms that are off-equilibrium with respect to net force via $w_i$. Detailed descriptions of other metrics are provided in Appendix C.2

**Baselines.** Following Xu et al. (2024), we compared our REBIND against traditional cheminformatic methods, DG and ETKDG algorithms, from RDKit (Landrum et al., 2013) and five representative GNNs. The GNNs considered in our experiments can be broadly broadly categorized as

(1) traditional GNNs such as GINE (Hu* et al., 2020), GATv2 (Brody et al., 2022), DeeperGCN-DAGNN adapted from Xu et al. (2021c), (2) graph transformers including GraphGPS (Rampasek et al., 2022) and GTMGC (Xu et al., 2024). For GraphGPS, we conducted experiments using both random walk (RW) and Laplacian (LP) positional encoding strategies. To ensure a fair comparison, we trained the MoleBERT (Xia et al., 2023) tokenizer in GTMGC separately for each dataset, which differs from the original setting where the tokenizer trained on a Molecule3D random split is used universally across all benchmarks. To further validate the efficacy of our framework, we included a comparison with the diffusion-based conformer generation model, Torsional Diffusion (Jing et al., 2022). Detailed experimental configurations are specified in Appendix C.

## 5.2 MAIN RESULTS

We present the performance results with of REBIND in Table 1, where the percentage reduction is calculated as the ratio of the performance improvement over the best baseline, expressed as a percentage. As demonstrated, our method achieves consistent superiority over baseline methods across all datasets and evaluation splits. On the QM9 dataset, REBIND achieves notable improvements across all evaluation metrics, with gains of up to 24% in both C-RMSD and E-RMSD. Additionally, our method exhibits robust generalization capabilities on the larger Molecule3D dataset. Notably, REBIND further excels in the scaffold split, where training and test sets contain structurally diverse molecules, achieving up to a 10% reduction in E-RMSD. This highlights REBIND's ability to generalize effectively to novel molecular scaffolds. Furthermore, it is worth emphasizing that higher performance gains are observed in E-RMSD compared to C-RMSD. This suggests that REBIND generates molecular conformations that are not only geometrically accurate but also more *realistic* in terms of energetic stability, resulting in more physically plausible molecular structures.

## 5.3 SCALABILITY TO LARGE MOLECULES

We further validate REBIND on the GEOM-DRUGS dataset with large molecules by including Torsional Diffusion (Jing et al., 2022)as an additional baseline, which adopts the same dataset as its benchmark. Since Torsional Diffusion was originally designed to generate multiple conformers, we adapt it to produce a single conformation per molecule for evaluation.

As shown in Table 2, REBIND significantly outperforms the baselines, achieving improvements of up to 4% in C-RMSD

Table 2: Conformer prediction performance and percentage reduction (%) of REBIND compared to the best baseline performance on the GEOM-DRUGS dataset.

| Splits | | Test | | | |
|---|---|---|---|---|---|
| **Datasets** | **Methods** | D-MAE ↓ | D-RMSE ↓ | C-RMSD ↓ | E-RMSD ↓ |
| | RDKit-DG | 1.181 | 2.132 | 2.097 | 3.623 |
| | RDKit-ETKDG | 1.120 | 2.055 | 1.934 | 3.330 |
| | GINE | 1.125 | 1.777 | 2.033 | 3.925 |
| | GATv2 | 1.042 | 1.662 | 1.901 | 3.728 |
| **GEOM-DRUGS** | GraphGPS (RW) | 0.879 | 1.399 | 1.768 | 3.472 |
| | GraphGPS (LP) | 0.815 | 1.300 | 1.698 | 3.171 |
| | GTMGC | 0.823 | 1.319 | 1.458 | 2.830 |
| | Torsional Diffusion | 0.959 | 1.648 | 1.751 | 2.992 |
| | REBIND | **0.776** | **1.283** | **1.396** | **2.602** |
| | Percentage Reduction ↑ | 4.79 | 1.31 | 4.25 | 8.06 |

and 8% in E-RMSD. Diffusion-based models like Torsional Diffusion are designed to generate multiple stable conformations and are not directly suited for predicting the single most stable conformation of a molecule, resulting in suboptimal performance. In contrast, REBIND demonstrates strong generalizability to drug-like molecules, surpassing diffusion-based models with lower computational costs and eliminating the need for multiple inference steps.

## 5.4 ABLATIVE STUDY

**Ablation on components.** We conduct an ablation study on the components of REBIND on the QM9 dataset, with the results presented in Table 3. Using a fully-connected transformer with Laplacian positional embedding $L$ results in limited performance, highlighting that uniformly

Table 3: Ablative study of REBIND.

| $L$ | $A^{bond}$ | $A^{near}$ | $\mathcal{A}^{force}$ | C-RMSD ↓ | E-RMSD ↓ |
|---|---|---|---|---|---|
| ✓ | | | | 0.5294 | 1.0323 |
| ✓ | ✓ | | | 0.3524 | 0.6920 |
| ✓ | ✓ | ✓ | | 0.3522 | 0.6807 |
| ✓ | ✓ | | ✓ | **0.3209** | **0.6103** |

connecting edges without accounting for inter-atomic forces is suboptimal. Integrating the bond-based adjacency matrix $A^{bond}$ with $L$ leads to substantial performance improvements. Moreover, we evaluate the impact of incorporating force-aware edges (*i.e.*, $\mathcal{A}^{force}$) compared to edge augmen-

tation based on proximity (*i.e.*, $\boldsymbol{A}^{\text{near}}$), the strategy adopted in Luo et al. (2021). The distance-based edges $\boldsymbol{A}^{\text{near}}$ are constructed using the intermediate distance matrix $\widehat{\boldsymbol{D}}^{\text{enc}}$, connecting atoms $i$ and $j$ when $\widehat{\boldsymbol{D}}_{ij}^{\text{enc}} < \delta$. The threshold $\delta$ is set as $10\,\text{Å}$, following the protocol in Luo et al. (2021). Our results demonstrate that $\mathcal{A}^{\text{force}}$, which accounts for significant inter-atomic forces influencing spatial conformation, consistently outperforms $\boldsymbol{A}^{\text{near}}$, showing up to 9% improvements in C-RMSD and larger gains up to 10% in E-RMSD. This demonstrates the effectiveness of $\mathcal{A}^{\text{force}}$ in aiding the model to generate conformations that are both closer to the ground state and energetically stable.

**Application to GNN-based architectures.** The core concept of REBIND is adaptable to any encoder-decoder framework. To demonstrate its versatility, we integrate REBIND with three GNN-based architectures: GINE, GATv2, and GraphGPS (LP). We implement REBIND by dividing the $L$ layers of original GNNs into two halves, where the top half operates as the encoder and the bottom half operates as the decoder. To compute $\mathcal{A}^{\text{force}}$, we follow the same procedure as REBIND, wherein the encoder's output is passed through a task-specific head layer to generate $\widehat{\boldsymbol{C}}^{\text{enc}}$ and $\widehat{\boldsymbol{D}}^{\text{enc}}$.

The results are shown in Table 4 of Appendix B. Results demonstrate that incorporating REBIND significantly improves performance across all backbone architectures and metrics, with gains of at least 5%. Notably, when applied to GraphGPS, we achieve the lowest prediction errors, with improvements of 5.11% in C-RMSD and 8.74% in E-RMSD. These results underscore the versatility of REBIND in boosting performance regardless of the underlying architecture. Moreover, REBIND's integration requires minimal implementation effort, making it compatible with any existing GNN-based architecture through the proposed mechanism.

## 5.5 QUALITATIVE ANALYSIS

We present a qualitative comparison of REBIND against baseline architectures based on atom-wise RMSD on the QM9 dataset, shown as blue bars in Figure 1. As illustrated, REBIND achieves significant error reductions for both groups. Specifically, when compared against GTMGC, REBIND further reduces the atom-wise RMSD for lower-degree groups by 0.11, compared to a reduction of 0.05 for high-degree groups, thereby verifying the efficacy of our degree-compensating augmentation. Moreover, our method greatly reduces the gap between the atom-wise errors of low-degree and high-degree groups, achieving 15.16% and 20.91% percentage reduction in the gap for atom-wise RMSD and MAE, respectively, compared to GTMGC. These results substantiates that REBIND comprehensively captures non-bonded interactions through force-aware edge augmentation.

## 6 RELATED WORKS

There has been a growing interest in generating molecular conformations from 2D molecular graphs. Traditional cheminformatics methods, such as DG and ETKDG from RDKit (Landrum et al., 2013), are widely used for their efficiency, relying on chemical heuristics for fast generation. However, their performance is limited due to insufficient handling of non-bonded interactions and minimal energy optimization. To address this limitation, deep learning models have emerged as powerful alternatives, producing multiple conformations for molecules using generative models (Mansimov et al., 2019; Xu et al., 2021b;a; Luo et al., 2021; Shi et al., 2021b; Xu et al., 2022; Jing et al., 2022). Recently, the focus has shifted to predicting ground-state molecular conformations, which is critical for practical applications requiring stability and feasibility. A benchmark and training pipeline were introduced in Xu et al. (2021c) for this task, followed by the graph transformer (Xu et al., 2024), achieving state-of-the-art results. Further details on related works are provided in Appendix A.

## 7 CONCLUSION

Given the limitations of previous studies in modeling inter-atomic forces, we introduced REBIND, an innovative framework that incorporates force-aware edge augmentation to accurately capture inter-atomic interactions. We anticipate that REBIND can be extended to other critical applications such as drug discovery or property prediction, particularly in scenarios where ground-truth coordinates are unavailable. As a future work, we plan to integrate our framework with these applications to enhance their predictive accuracy.

ACKNOWLEDGMENTS

This work was supported by Institute for Information & communications Technology Planning & Evaluation(IITP) grant funded by the Korea government(MSIT) (RS-2019-II190075, Artificial Intelligence Graduate School Program(KAIST)). We sincerely appreciate Polymerize for their generous support. Special thanks also go to Juneyong Yang and Changhun Kim for invaluable comments in improving this manuscript.

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

## SUPPLEMENTARY MATERIALS

Table 4: Conformer prediction performance and percentage reduction (%) of GNN-based backbones integrated with REBIND on the QM9 dataset.

| Method | | Test | | | |
|---|---|---|---|---|---|
| Backbone | Methods | D-MAE ↓ | D-RMSE ↓ | C-RMSD ↓ | E-RMSD ↓ |
| **GINE** | Vanilla | 0.606 | 0.946 | 0.867 | 1.696 |
| | + REBIND | **0.505** | **0.761** | **0.801** | **1.605** |
| | Reduction ↑ | 16.65 | 19.55 | 7.61 | 5.36 |
| **GATv2** | Vanilla | 0.382 | 0.690 | 0.712 | 1.358 |
| | + REBIND | **0.319** | **0.523** | **0.611** | **1.198** |
| | Reduction ↑ | 16.62 | 24.14 | 14.14 | 11.77 |
| **GraphGPS (LP)** | Vanilla | 0.283 | 0.499 | 0.546 | 1.064 |
| | + REBIND | **0.211** | **0.361** | **0.518** | **0.971** |
| | Reduction ↑ | 25.55 | 27.58 | 5.11 | 8.74 |

## A RELATED WORKS

**Molecular Conformer Generation.** In recent years, there has been significant progress in the field of molecular conformation generation from 2D molecular graphs. Traditional cheminformatics approaches, such as Distance Geometry (DG) and its extension, ETKDG (Landrum et al., 2013), have been widely employed due to their computational efficiency. These methods rely on chemical heuristics and geometric rules to generate conformations efficiently, which makes them attractive for large-scale applications. However, their reliance on approximations and minimal energy optimization limits their ability to accurately capture non-bonded interactions, which play a crucial role in determining the stability and realism of molecular conformations.

To overcome the limitations of traditional approaches, deep learning models have emerged as promising choices. Generative models, in particular, have been extensively explored for molecular conformation generation, offering more accurate and realistic predictions. Early works (Mansimov et al., 2019; Xu et al., 2021b) leverage variational autoencoders (VAEs) to encode molecular information into latent spaces, where each molecule's structure is represented as a probabilistic distribution. From this latent space, multiple plausible 3D conformations are sampled by decoding the latent variables. Similarly, flow-based models (Xu et al., 2021a) generate molecular conformations by transforming latent variables sampled from a Gaussian distribution into atomic distance matrices, capturing long-range dependencies between atoms. The 3D coordinates are then derived from the generated distances, followed by refinement using energy-based models. Score-based models have also gained attention for molecular conformation generation. These models learn a score function representing the gradient of the log probability density of atomic coordinates (Luo et al., 2021; Shi et al., 2021b). By perturbing the molecular data with Gaussian noise at multiple levels, these models iteratively denoise the data using the learned score function to guide the generation of valid conformations. Building on the success of diffusion models (Ho et al., 2020) in the vision domain, their application to molecular conformation generation has become another emerging area of interest (Xu et al., 2022; Jing et al., 2022; Zhang et al., 2023; Fan et al., 2024). They have shown remarkable success in generating diverse molecular geometries by learning the desired geometric distribution from a noise distribution through a reverse diffusion process.

**Ground-state molecular conformer prediction.** While the one-to-many task of generating multiple conformations has been thoroughly studied, recent research has shifted toward the prediction of a molecule's ground-state conformation. This focus reflects the practical importance of identifying the most stable and energetically favorable molecular structure, which is essential for real-world applications such as drug design and material discovery. Ground-state conformation prediction requires models to identify the most stable configuration on the molecule's potential energy surface, which is challenging for one-to-many methods to address.

To facilitate progress in this area, the Molecule3D benchmark and training pipeline are introduced in Xu et al. (2021c). They provide a standardized dataset and evaluation protocols for predicting a molecule's ground-state geometry. Building on this, the graph transformer architecture (GT-MGC) (Xu et al., 2024) was recently proposed, tailored to predict ground-state molecular confor-

mations. GTMGC incorporates bonds and pairwise distances within the self-attention mechanism to capture both local and global molecular interactions, achieving state-of-the-art performance on the benchmarks with ground-state conformations.

Table 5: Examples of visualizations and RMSD predictions from RDKit-ETKDG, GTMGC, and REBIND on the QM9 and GEOM-DRUGS datasets.

| Datasets | QM9 | | | GEOM-DRUGS | | |
|---|---|---|---|---|---|---|
| Ground-truth | | | | | | |
| RDKit-ETKDG | 1.233 | 0.844 | 1.621 | 3.033 | 2.515 | 3.470 |
| GTMGC | 1.242 | 1.168 | 1.479 | 2.295 | 2.232 | 2.521 |
| REBIND | **0.050** | **0.058** | **0.376** | **0.844** | **0.898** | **1.174** |

## B  ADDITIONAL EXPERIMENTS

### B.1  MOLECULAR VISUALIZATIONS

In this section, we provide 3D visulaizations and corresponding RMSD predictions from our method, RDKit-ETKDG, and GTMGC on the QM9 and GEOM-DRUGS dataset. As illustrated, the conformations predicted by REBIND shows significantly closer alignment with the ground-truth conformations. This superiority is consistent across molecules of varying sizes, demonstrating the robustness of our approach.

### B.2  FURTHER QUALITATIVE ANALYSIS

We present an additional qualitative comparison of REBIND against baseline architectures based on atom-wise E-RMSD, as shown in Figure 1. Analogous to previous analyses, we calculated the atom-wise E-RMSD between the ground-truth 3D coordinates and predictions on the QM9 dataset. For an atom $i$ in the $m$-th molecule, the atom-wise E-RMSD, E-RMSD$(i)$, is formulated as:

$$\text{E-RMSD}(i) = \frac{p^{(m)}}{\widehat{p}^{(m)}} w_i \|\widehat{C}_i - C_i\|_2, \quad (16)$$

where $\widehat{C}_i$ and $C_i$ denotes the predicted and ground-truth coordinates of atom $i$. $\frac{p^{(m)}}{\widehat{p}^{(m)}}$ is a Boltzmann factor of the $m$-th molecule and $w_i$ denotes the normalized atom-wise force, as de-

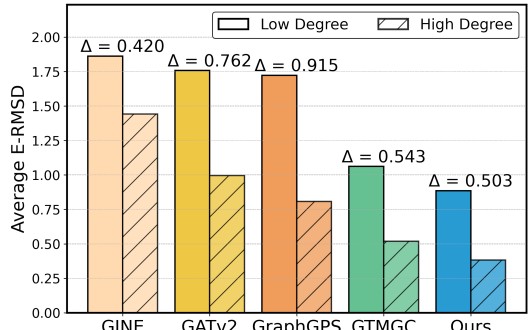

Figure 4: Atom-wise E-RMSD analysis with respect to the relative atom degree on REBIND and baselines. $\Delta$ denotes the gap between the errors of the low-degree and high-degree bins.

tailed in Section 3. Following the same procedure, we categorized nodes into the low-degree and high-degree groups based on the relative degree, and computed the average atom-wise RMSD for each group, as depicted in Figure 4.

Consistent with the results in Section 5.5, REBIND achieves significant error reductions for both low-degree and high-degree groups. Specifically, when compared against GTMGC, REBIND further reduces the atom-wise E-RMSD for lower-degree groups by 0.18, compared to a reduction of 0.14 for high-degree groups. Furthermore, our method greatly reduces the gap between the atom-wise

Table 6: Comparison of predicted molecules' energy from models across datasets, denoted in KJ · mol$^{-1}$. Best results are shown in **bold**, while last column denotes percentage reduction with respect to the second best performance.

| Datasets | GAT | GINE | GraphGPS | GTMGC | Ours | % Reduction |
|---|---|---|---|---|---|---|
| QM9 | 3.28 | 9.26 | 0.38 | 0.39 | **0.26** | 33.3 |
| Scaffold | 91.20 | 131.30 | 0.73 | 0.56 | **0.32** | 42.6 |
| Random | 10.40 | 5.50 | 1.38 | 0.56 | **0.27** | 51.8 |
| GEOM-Drugs | 271.86 | 270.00 | 195.17 | 1.75 | **1.34** | 23.4 |

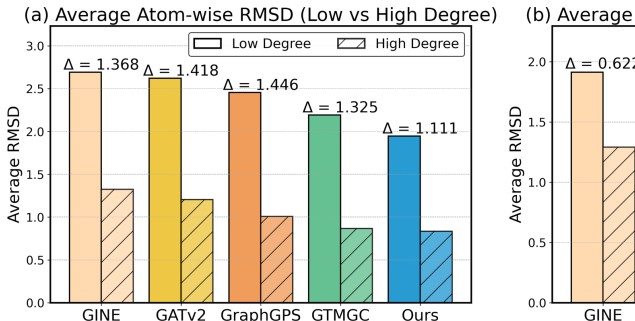

Figure 5: Further analysis of Atom-wise RMSD on (a) GEOM-DRUGS and (b) Molecule3D datasets. $\Delta$ denotes the gap between the errors of the low-degree and high-degree groups.

RMSD of low-degree and high-degree groups, achieving a 7.37% percentage reduction in the gap compared to GTMGC, demonstrating the efficacy of REBIND in accurately modeling interatomic interactions.

### B.3 IMPORTANCE OF LOW-DEGREE ATOMS

To highlight the performance disparities between low-degree and high-degree atoms discussed in Section 3, we further visualize the proportion of low-degree atoms within each molecule in Figure 6, using the Molecule3D dataset. The visualization reveals that the proportion of low-degree atoms ($\deg_i^{\text{rel}} \in (0, 0.3]$) increases significantly as molecular size grows. Combined with our findings in Section 3, this suggests that inaccuracies in predicting the positions of low-degree atoms are a major contributor to performance degradation in conformation prediction for large molecules. These observations emphasize the necessity of novel methods, such as REBIND, to address the challenges associated with low-degree atoms effectively.

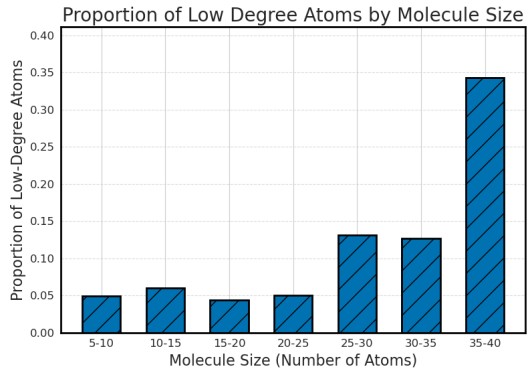

Figure 6: Proportion of low-degree ($\deg_i^{\text{rel}} \in (0, 0.3]$) within each atom, binned with respect to their molecular sizes. Molecule3D is used during visualization.

### B.4 PREDICTION OF LOW-ENERGY CONFORMATIONS

To highlight REBIND's capability in predicting low-energy (*i.e.*, stable) conformations, we conduct an energy-based validation experiment. Specifically, we compute the average energy difference between the predicted conformations and the ground-state conformations across all baseline methods. The energy calculations were performed using the Merck Molecular Force Field (MMFF) in RD-Kit. The results are shown in Table 6, demonstrating that REBIND consistently achieves the lowest energy differences ($|E_{\text{pred}} - E_{\text{ground}}|$) compared to baseline methods. This finding underscores RE-

Table 7: Comparison of time complexity among GTMGC and REBIND. The experiments are conducted on the machine with NVIDIA GeForce RTX 3090 GPU/Intel(R) Xeon(R) Gold 5215 CPU @2.50GHz.

| Time (sec.) | QM9 | | | Molecule3D | | | Geom-DRUGS | | |
|---|---|---|---|---|---|---|---|---|---|
| | Forward | Backward | Total | Forward | Backward | Total | Forward | Backward | Total |
| $A^{\text{bond}}$ | 0.029 | 0.038 | 0.067 | 0.030 | 0.037 | 0.067 | 0.094 | 0.105 | 0.199 |
| **GTMGC** | 0.031 | 0.039 | 0.070 | 0.031 | 0.047 | 0.077 | 0.129 | 0.144 | 0.273 |
| $A^{\text{bond}} + A^{\text{near}}$ | 0.059 | 0.040 | 0.010 | 0.061 | 0.043 | 0.104 | 0.169 | 0.092 | 0.261 |
| **Ours** | 0.064 | 0.043 | 0.107 | 0.069 | 0.041 | 0.110 | 0.195 | 0.109 | 0.304 |

BIND's ability to reliably predict the most stable conformation, which aligns closely with our task objective of identifying the global energy minimum.

## B.5 COMPUTATION BURDEN ASSESSMENT

For rewiring the edges incurs additional computational overhead, we provide a detailed comparison of both forward and backward computation times (wall-clock time) across all three datasets. Specifically, we compare the time complexity between 1. incorporating $A^{\text{bond}}$ alone, 2. $A^{\text{bond}} + D^{\text{row-sub}}$ (*i.e.*, GTMGC (Xu et al., 2024)), 3. $A^{\text{bond}} + A^{\text{near}}$ (*i.e.*, adjacency matrix constructed via close distance), and 4. REBIND. The results are shown in Table 7. Note that, the results are averaged upon 50 gradient update steps for each dataset, for they have varying molecular sizes (8, 15, and 25 average molecular size for each dataset). As expected, the computation time increases from QM9 to GEOM-DRUGS due to the larger average molecular sizes. On average, our method introduces an additional 0.03 seconds of computational overhead compared to GTMGC, due to the calculation of the Lennard-Jones (LJ) potential matrix.

## B.6 ANALYSIS UPON VARIOUS REWIRING TECHNIQUES

To further provide insights upon edge-rewiring of edges, we conduct a extensive analysis upon the combinations of possible construction of adjacency matrices. Specifically, (1) a comparison between $\mathcal{A}^{\text{force}}$ and $A^{\text{near}}$ in the absence of $A^{\text{bond}}$ and $D^{\text{row-sub}}$ and (2) additional combinations of these four components. As shown in Table 8, incorporating our proposed $\mathcal{A}^{\text{force}}$ to $A^{\text{bond}}$ and $D^{\text{row-sub}}$ significantly reduces errors, outperforming those involving $A^{\text{near}}$ and $D^{\text{row-sub}} + A^{\text{bond}}$ setting. It is important to note that REBIND is designed to integrate distance and force information in a complementary manner. As such, employing $\mathcal{A}^{\text{force}}$ alone may yield limited performance improvements due to the absence of distance factors (*e.g.*, atomic pairs with close distances but repulsive forces). However, when

Table 8: Comparison of possible combinations of rewiring techniques in QM9 dataset.

| Combinations | C-RMSD↓ | E-RMSD↓ |
|---|---|---|
| Vanilla Transformer | 0.529 | 1.032 |
| $\mathcal{A}^{\text{force}}$ | 0.483 | 0.951 |
| $A^{\text{near}}$ | 0.454 | 0.903 |
| $D^{\text{row-sub}} + \mathcal{A}^{\text{force}}$ | 0.328 | 0.646 |
| $A^{\text{bond}} + \mathcal{A}^{\text{force}}$ | 0.359 | 0.712 |
| $A^{\text{near}} + A^{\text{bond}}$ | 0.337 | 0.655 |
| $A^{\text{bond}} + D^{\text{row-sub}} + A^{\text{near}}$ | 0.349 | 0.694 |
| $A^{\text{bond}} + D^{\text{row-sub}} + \mathcal{A}^{\text{force}}$ | **0.326** | **0.617** |

combining interatomic forces with $A^{\text{near}}$ or $D^{\text{row-sub}}$, it achieves substantial improvements in conformation prediction accuracy by effectively modeling both interatomic forces and spatial relationships.

## C EXPERIMENTAL SETTINGS

### C.1 DATASET DESCRIPTION

We evaluate our method on three benchmark datasets: QM9, Molecule3D, and GEOM-Drugs. Details for each datasets are specified below.

- **QM9** is a widely-used quantum chemistry dataset containing molecular geometries, electronic properties, and energy attributes for small organic molecules with up to 9 heavy atoms. The 3D conformations are obtained using density functional theory (DFT). We adopt the data split proposed by (Liao & Smidt, 2023).

- **Molecule3D** is a large-scale dataset consisting of approximately 4 million molecules. Each molecule is annotated with 2D molecular graphs, ground-state 3D conformations, and various quantum properties. We follow the splits used in (Xu et al., 2024), including a random split and a scaffold split. The scaffold split groups molecules based on their core substructures, allowing for a more realistic evaluation.
- **GEOM-Drugs** is a subset of the GEOM dataset, focusing on drug-like molecules. For each molecule, multiple conformation sets along with their chemical properties are provided. Thus, we choose the most stable conformation with respect to Boltzmann energy for each molecule for our experiments. For data splits, we adhere to the splits used in (Ganea et al., 2021).

## C.2    METRICS DESCRIPTION

We consider the following metrics for evaluation.

- **MAE**. Mean absolute Error (MAE) quantify the accuracy of predicted interatomic distances relative to the ground truth on a pairwise basis. Let $\widehat{\boldsymbol{D}}_{ij}$ and $\boldsymbol{D}_{ij}$ represent the predicted and ground truth distances between atoms $i$ and $j$, respectively. The metric is formulated as follows:

$$\text{MAE}(\widehat{\boldsymbol{D}}, \boldsymbol{D}) = \frac{1}{N^2} \sum_{i,j \in \mathcal{V}} |\widehat{\boldsymbol{D}}_{ij} - \boldsymbol{D}_{ij}|$$

- **RMSE**. Root Mean Square Error (RMSE), also quantifies the error in interatomic distances, on a pairwise basis. Specifically, the metric is formulated as follows:

$$\text{RMSE}(\widehat{\boldsymbol{D}}, \boldsymbol{D}) = \sqrt{\frac{1}{N^2} \sum_{i,j \in \mathcal{V}} (\widehat{\boldsymbol{D}}_{ij} - \boldsymbol{D}_{ij})^2}$$

- **RMSD**. Root Mean Square Deviation measures the spatial deviation between two conformations that are aligned using the Kabsch algorithm (Kabsch, 1978). Only heavy atoms (*i.e.*, excluding hydrogens) are considered during this calculation. Let $\widehat{\boldsymbol{G}}_i$ and $\boldsymbol{G}_i$ represent the aligned conformations of atom $i$. RMSD is defined as:

$$\text{RMSD}(\widehat{\boldsymbol{G}}, \boldsymbol{G}) = \sqrt{\frac{1}{N} \sum_{i \in \mathcal{V}} \|\widehat{\boldsymbol{G}}_i - \boldsymbol{G}_i\|_2}$$

- **E-RMSD**. Energy-weighted RMSD, considers chemical feasibility of the predictions on top of the spatial deviations. The probability likelihood of the whole preicted conformation $\widehat{\boldsymbol{G}}$ with respect to the ground state $\boldsymbol{G}$ is accounted via the Boltzmann factor $(i.e. \frac{\widehat{p}}{p} = \exp\left(\frac{\hat{E}-E}{kT}\right))$, while the feasibility at atom-level is considered via sum-normalized force $(i.e. w_i = \frac{F_i}{\sum_{j \in \mathcal{V}} F_j})$ acting upon each atom. Both the Boltzmann factor and force is calculated using Merck Molecular Force Field (Halgren, 1996). For all our experiments, we set $T = 298.15$ K and $k = 0.001987$ kcal mol$^{-1}$K$^{-1}$, reflecting standard laboratory conditions.

## C.3    IMPLEMENTATION DETAILS

We adopted the evaluation protocols and train/validation/test splits from Xu et al. (2024) for the QM9 and Molecule3D datasets and from Jing et al. (2022) for the GEOM-DRUGS dataset. Test results were derived from the best-performing model on the validation set based on the D-MAE metric. All GNN architectures were implemented using PyTorch Paszke et al. (2019) and PyTorch Geometric Fey & Lenssen (2019). The experiments were conducted on RTX Titan and RTX 3090 (24GB) GPU machines. Throughout all experiments except for Jing et al. (2022); Xu et al. (2024), we set the hidden dimension to 512 and the number of layers to 8; for these references, we adopted their optimal configurations. We employed the AdamW optimizer with a batch size of 100 and no weight decay. Learning rates were initially warmed up from 0 and then fixed based on the best validation performance on the QM9 dataset, within the range of [3e-5, 5e-5, 7e-5, 9e-5]. The number of attention heads was set to 8. We used a seed of 42 for all experiments and trained all models for 20 epochs, following the configuration in Xu et al. (2024).

