# OpenReview forum: "REBIND: Enhancing Ground-state Molecular Conformation Prediction via Force-Based Graph Rewiring"
_ICLR.cc/2025/Conference — ICLR 2025 Poster_

### Official Review · Reviewer_twMi · 2024-10-28

**Soundness:** 3
**Presentation:** 3
**Contribution:** 3
**Rating:** 8
**Confidence:** 4

**Summary:**

This article proposes a method, REBIND, to enhance the performance of deep learning (DL)-based molecular ground-state conformation prediction. The authors begin by identifying a key limitation in current approaches, which primarily rely on chemical bond information and interatomic distances but fail to adequately model atomic interactions, particularly for non-bonded atom pairs influenced by van der Waals potentials. Through detailed observations and statistical analysis, the authors conclude that current methods tend to accumulate more significant errors in predicting conformations of low-degree (i.e., low coordination numbers) molecules with fewer chemical bonds. This supports their viewpoint on the limitations of existing methods. To address this shortcoming, the authors introduce an approach based on the Lennard-Jones potential, deriving a new connectivity framework (comprising two adjacency matrices, $A^{attr}$ for attractive forces and for $A^{rep}$ repulsive forces) to redefine the molecular graph. By adding these new edges, the model enhances the representation of non-bonded atomic interactions as a form of data augmentation. The overall model architecture builds upon the GMTGC [1] network, replacing its original interatomic relation modeling component, $D^{row-sub}$, with the newly proposed $A^{attr}$ and $A^{rep}$, thereby strengthening the native self-attention mechanism. The results are clear and well-documented, strongly supporting the authors’ assertions. Comparative experiments demonstrate that these improvements lead to significant performance gains across various datasets.

For me, a major strength of this work is its clear observations and straightforward improvements that lead to noticeable performance gains, with a well-organized structure throughout. However, the methodological contributions feel somewhat modest. Given that molecular ground-state conformation prediction is still an underexplored area, this work holds potential to drive further advancements in the field.

[1] Xu, Guikun, et al. "GTMGC: Using Graph Transformer to Predict Molecule’s Ground-State Conformation." *The Twelfth International Conference on Learning Representations*.

**Strengths:**

1. The article is well-structured, with a clear and logical flow.

2.	The authors begin by hypothesizing and questioning the accuracy of existing methods for modeling atomic interactions, then validate their assumptions through experimental observations. Building on this, they propose an improved method, making the work both robust and effective.

3. The newly proposed metric, Energy-weighted RMSD (E-RMSD), is a practical contribution, enhancing evaluation standards for this task within the research community.

4. The extension of experiments to the GEOM-DRUGS dataset is also a valuable contribution, enriching the research standards for this field.

5.	The comparison in the ablation study between $A^{near}$ and $A^{force}$ effectively demonstrates the advantage of the authors’ proposed edge-enhancement method over the traditional distance-threshold-based neighbor edge enhancement. This approach could potentially be applied as a new edge construction strategy in other molecular modeling fields.

**Weaknesses:**

1. Some claims in the paper are unclear:
   1. In the Introduction, the statement “To this end, we propose a novel encoder-decoder graph transformer architecture…” is ambiguous. The process of using a Transformer encoder to predict the initial conformation, followed by a decoder to refine the final conformation, is actually a contribution from the GTMGC [1] architecture and should be properly cited. The main contribution of this paper is the improvement of how atomic interactions are modeled within the GTMGC framework.
   2. The definition of ${deg}^{rel}_i$ should be included and thoroughly explained in the Notation section, as it is the key metric used to classify nodes into low-degree and high-degree categories.

2. The ablation study would be more convincing if it included an analysis of the effects of removing $A^{bond}$ and$D^{row-sub}$ . Additionally, testing combinations of $A^{bond}$, $D^{row-sub}$ , $A^{near}$ and $A^{force}$ could reveal whether certain configurations lead to better performance.

3. In my opinion, the subjective evaluation results (Table 5) should be included in the main body of the paper for better clarity.

**Questions:**

1. I would like to know the results of the observations in **Figure 1** on large datasets like Molecule3D.
2. I would like to understand the trade-off in terms of overall forward pass time cost after incorporating compared to GTMGC.
3. **as Weaknesses.2**

---

> ### Author Response · Authors · 2024-11-21
> **Response to Reviewer twMi (1)**
>
> **W1. Revision of Manuscript**
>
> We thank the reviewer for the comments in improving our manuscript. We have revised our manuscript, addressing your concerns.
>
> Specifically:
> - The statement in the Introduction section is modified as below:
> > To this end, we propose a force-aware self-guidance graph transformer, whereupon the initial conformation predicted from the encoder, a LJ potential-based adjacency matrix is constructed, then utilized in the decoder to predict the final conformation.
> - We added the definition of the relative degree in the preliminary section, for clarity.
>
> Additionally, following the reviewer’s suggestion, we will replace Table 5 to our main paper in the subsequent version of our manuscript.
>
> **W2 & Q3. Further Ablative Studies**
>
> In response to the reviewer’s helpful suggestion, we have extended our ablative studies to include: (1) a comparison between $A_{force}$ and $A_{near}$ in the absence of $A^{bond}$ and $D^{row-sub}$ and (2) additional combinations of these four components. As shown in the table below, incorporating our proposed $A_{force}$ to $A^{bond}$ and $D^{row-sub}$ significantly reduces errors, outperforming those involving $A_{near}$ and $D^{row-sub}$+$A^{bond}$ setting. It is important to note that REBIND is designed to integrate distance and force information in a complementary manner. As such, employing $A_{force}$ alone may yield limited performance improvements due to the absence of distance factors (*e.g.*, atomic pairs with close distances but repulsive forces). However, when combining interatomic forces with $A_{near}$ or $D^{row-sub}$, it achieves substantial improvements in conformation prediction accuracy by effectively modeling both interatomic forces and spatial relationships.
>
> Table 1. Further ablative study of REBIND on the QM9 dataset.
> | Combinations                             |   C-RMSD   |  E-RMSD  |
> |------------------------------------------|------------|----------|
> | Only Transformer                         |   0.5294   |  1.0323  |
> | $A_{force}$                              |   0.4826   |  0.9510  |
> | $A_{near}$                               |   0.4541   |  0.9033  |
> | $D^{row-sub}$ + $A_{force}$              |   0.3284   |  0.6457  |
> | $D^{row-sub}$ + $A_{near}$               |   0.3588   |  0.7118  |
> | $D^{row-sub}$ + $A^{bond}$               |   0.3466   |  0.6552  |
> | $A^{bond}$ + $D^{row-sub}$ + $A_{force}$ |   **0.3260**   |  **0.6165**  |
> | $A^{bond}$ + $D^{row-sub}$ + $A_{near}$  |   0.3489   |  0.6937  |

---

> > ### Author Response · Authors · 2024-11-21
> > **Response to Reviewer twMi (2)**
> >
> > **Q1. Additional Observation on Large Datasets**
> >
> > We present the visualization results of atom-wise RMSD for each group of low-degree and high-degree atoms on (a) GEOM-Drugs and (b) Molecule3D in Figure 5 in Appendix of our revised manuscript. Consistent with the observations on QM9, we observe significant gaps between low-degree and high degree atom’s conformation errors. Furthermore, REBIND demonstrates state-of-the art performance in mitigating the performance discrepancy; showing on average 20% decrease of discrepancy between low-degree and high-degree atoms’ performance.

---

> ### Author Response · Authors · 2024-11-21
> **Response to Reviewer twMi (3)**
>
> **Q2. Computational Burden Assessment**
>
> We appreciate the reviewer's insightful comment. To address your concerns, we provide a detailed analysis of both forward and backward computation times (wall-clock time) across all three datasets. Specifically, we compare the time complexity of the following configurations: (1) Utilizing a sole $A^{bond}$, (2) GTMGC [1] that incorporates $A^{bond}$ + $D^{row-sub}$, and (3) REBIND ($A^{bond}$ + $A_{force}$). The results are averaged over 50 gradient steps and calculated separately for each dataset, as they exhibit varying average molecular sizes (8, 15, and 25 for QM9, Molecule3D, and GEOM-DRUGS, respectively).
>
> As expected, the computation time increases from QM9 to GEOM-DRUGS due to the larger average molecular sizes. On average, our method introduces an additional 0.03 seconds of computational overhead compared to GTMGC, due to the calculation of the Lennard-Jones (LJ) potential matrix. For clarity, the summarized results are provided below.
>
> Table 2. Computation time analysis of REBIND (Ours) and baselines. The experiments are conducted on the machine with NVIDIA GeForce RTX 3090 GPU/Intel(R) Xeon(R) Gold 5215 CPU @2.50GHz.
> |           Sec.             | QM9     |      |       | Molecule3D  |     |    | Geom-DRUGS  |     |    |
> |------|------|---------|---------|-------|-------|--------|--------|---------|------|
> |                        | forward | backward | total   | forward | backward | total   | forward | backward | total   |
> | $A^{bond}$                | 0.0287  | 0.0379   | 0.0666  | 0.0298  | 0.0367   | 0.0665  | 0.0943  | 0.105    | 0.1993  |
> | **GTMGC**        | 0.0311  | 0.039    | 0.0701  | 0.0307  | 0.0467   | 0.0774  | 0.12888 | 0.1442   | 0.27308 |
> | **Ours**                  | 0.0637  | 0.0428   | 0.1065  | 0.0691  | 0.0412   | 0.1103  | 0.1947  | 0.109    | 0.3037  |
>
> ---
> **References**
>
> [1] GTMGC: Using Graph Transformer to Predict Molecule’s Ground-State Conformation, 24’ ICLR

---

> > ### Comment · Reviewer_twMi · 2024-11-23
> >
> > Thank you to the authors for their response and hard work!
> >
> > Based on the authors’ detailed replies and additional experimental results, I have decided to raise my score for the following reasons:
> >
> > 1. The authors have effectively addressed the weaknesses I previously pointed out in the earlier version, including **contribution claims**, **necessary definitions in the notation section**, and **more ablation studies**.
> >
> > 2. The newly added **Figure 5**, in conjunction with **Figure 1** from the main text, provides a thorough resolution to my concerns. It convincingly demonstrates that their observation—**current methods tend to accumulate more significant errors in predicting the conformations of low-degree (i.e., low coordination numbers) atoms or molecules with fewer chemical bonds**—holds consistently across both small-molecule datasets (QM9) and large-molecule datasets (Molecule3D, Drugs). This strongly supports the motivation of their work.
> >
> > 3. The newly added **Computational Burden** experiment is presented clearly. I find the additional computational cost to be acceptable, especially given the significant performance improvements demonstrated by the proposed method.

---

> ### Author Response · Authors · 2024-11-23
>
> We sincerely thank the reviewer for the reassessment and positive opinion for our work! We are extremely glad that our responses have met and addressed your concerns.

---

### Official Review · Reviewer_8Kvn · 2024-11-01

**Soundness:** 3
**Presentation:** 4
**Contribution:** 2
**Rating:** 6
**Confidence:** 3

**Summary:**

This paper proposes a method to enhance the conformation prediction for non-bonded edges, inspired by the Lennard-Jones potential. Experimental results show that their method can improve the performance across multiple datasets, especially for those atoms with lower ranks.

**Strengths:**

In general, this paper is a well-written ML paper.

1. The background and motivation are clarified clearly. The methodology is reasonable and well-aligned with the motivation.

2. The empirical study shows a consistent improvement on multiple datasets.

**Weaknesses:**

My main concern is whether the "large errors" observed for low-degree atoms arise from methodological limitations or issues in the evaluation process. Additionally, it's essential to assess how significant these "errors" might be for practical applications.

There might be multiple conformations with similar energy levels, thus they would probably occur with comparable Boltzmann weights. If this is the case, the sampling bias in the training/evaluation set might be the reason for large errors for low-degree atoms.
If a predicted conformation has a large RMSD with one ground truth, maybe it is also in an energy-favored state, but not included in the evaluation.

Furthermore, flexibility in non-bonded conformations may not adversely impact empirical applications, as real-world cases often involve ensembles of multiple conformations.

**Questions:**

(1) As mentioned in the "Weaknesses" section, it would be helpful if the authors could provide a detailed analysis of the large errors in non-bonded interactions. Specifically, clarifying whether these errors are primarily attributable to model limitations rather than to the incomplete representation of ground-truth conformations would strengthen the discussion.

(2) The "E-RMSD" metric combines normalized force and Boltzmann factors, which makes it challenging to distinguish the contributions of each. It would be beneficial to disentangle these components, perhaps by introducing Boltzmann factors/force alone.

---

> ### Author Response · Authors · 2024-11-21
> **Response to Reviewer 8Kvn (1)**
>
> **W1 & Q1. Regarding Problem Setting & Validity of Observations**
>
> We thank the reviewer for these thoughtful and constructive comments, which help improve the clarity of our manuscript. Below, we address the reviewer’s concerns: (1) We validate our observations, by clarifying our problem scenario and showing that there is no “sampling bias” for our task which focuses on predicting the most energetically stable conformation, and (2) Empirical verification that REBIND indeed predicts energetically stable conformations.
>
> **(1) Concern on Sampling Bias**
>
> We would like to clarify that the focus of REBIND is to predict the most stable molecular conformation, specifically the ground-state conformation, where the energy is at its global minimum. In all of our evaluation settings, only a single conformation exists as the ground truth — this is the conformation with the lowest possible energy, in-line with protein structure prediction [1, 2, 3, 4] and crystal structure prediction [5, 6, 7] tasks. Our task diverges from traditional molecular conformer generation tasks, which aim to generate a diverse set of low-energy conformations representing multiple local minima on the energy surface. In contrast, our work focuses solely on identifying the global minimum conformation.
> Consequently, our evaluation metrics are designed to align with this task objective, quantifying the prediction error relative to the ground-state conformation (the lowest-energy conformation). As such, we believe the “sampling bias” mentioned by the reviewer, which pertains to selecting one ground truth from an ensemble of conformations, does not apply to the context of our work. Since our problem is defined around a single ground-state conformation, this bias is inherently absent.
>
> **(2) Verification of Predicting “Low-Energy” Conformations**
>
> To further substantiate that REBIND is capable of predicting low-energy conformations, as aligned with its design, we conducted an energy-based validation experiment. Specifically, we computed the average energy difference between the predicted conformations and the ground-state conformations across all baseline methods. The energy calculations were performed using the Merck Molecular Force Field (MMFF) in RDKit.
>
> The results in Table 1 below demonstrate that REBIND consistently achieves the lowest energy differences ($|E_{\text{pred}} - E_{\text{ground}}|$) compared to the baseline methods. This finding underscores REBIND’s ability to reliably predict the most stable conformation, which aligns closely with our task objective of identifying the global energy minimum.
>
> Table 1. Energy difference of predicted conformations through different models from the ground-state.
> |$\text{KJ}\cdot\text{mol}^{-1}$|**QM9**|**Molecule3D (scaffold)**|**Molecule3D (random)**|**GEOM-DRUGS**|
> |---|---|---|---|---|
> |**GAT**|3.2842|91.2003|10.4040|271.8635|
> |**GINE**|9.2558|131.2986|5.5030|269.9960|
> |**GraphGPS**|0.3842|0.7318|1.3814|195.1740|
> |**GTMGC**|0.3882|0.5559|0.5557|1.7494|
> |**Ours**|**0.2632**|**0.3237**|**0.2736**|**1.3420**|
>
> ---
> **References**
>
> [1] Protein structure prediction with energy minimization and deep learning approaches, 23’ Natural Computing
>
> [2] Constructing effective energy functions for protein structure prediction through broadening attraction-basin and reverse Monte Carlo sampling, 19’ BMC Informatics
>
> [3] Rapid sampling of local minima in protein energy surface and effective reduction through a multi-objective filter, 13’ Proteome Science
>
> [4] Protein structure prediction by global optimization of a potential energy function, 99’ Proceedings of the National Academy of Sciences
>
> [5] Crystal Structure Prediction of Energetic Materials, 23’ Crystal Growth and Design
>
> [6] Minima Hopping Method for Predicting Complex Structures and Chemical Reaction Pathways, 18’ Handbook of Molecular Modeling
>
> [7] Crystal structure prediction accelerated by Bayesian optimization, 18’ Physics Review Materials

---

> ### Author Response · Authors · 2024-11-21
> **Response to Reviewer 8Kvn (2)**
>
> **W2. Upon the Importance of Ground-state Conformation**
>
> As stated by the reviewer, real-world cases often involve ensembles of multiple conformations. However, the ground-state conformation (the current target in which REBIND wishes to predict) is of paramount importance for its influence on physical, chemical and biological properties of molecules. For example, the ground-state conformation serves as the starting point for understanding molecular interactions in protein-ligand binding affinity. Misinterpretation of this conformation leads to errors in docking studies and binding energy calculations [1]. Furthermore, predicting the bulk properties of crystalline and polymeric materials, such as mechanical strength, optical properties, and conductivity, are rooted in the accurate spatial arrangement of molecules in their ground-state conformations. Averaging over ensembles may miss critical packing interactions or fail to capture key structural details [2].
>
> ---
> **References**
>
> [1] Leach et al., Molecular Modelling: Principles and Applications, 2001
>
> [2] Morsch et. al., Organic Chemistry, 2024 (compiled version from LibreTexts)

---

> > ### Author Response · Authors · 2024-11-21
> > **Response to Reviewer 8Kvn (3)**
> >
> > **Q2. On the Role of E-RMSD**
> >
> > We acknowledge the reviewer’s concern regarding the potential complexity of the **E-RMSD** metric and the challenge in isolating the contributions of the Boltzmann factor and normalized force. However, we believe that this combination is highly appropriate and effective for our current setup due to the **deterministic nature of our task**, which focuses on predicting a single, most stable (ground-state) conformation. In this deterministic setup, combining the Boltzmann factor and normalized force within E-RMSD allows us to evaluate both **energetic feasibility** and **geometric alignment** for the predicted structure relative to the ground truth. In essence, its main purpose is to distinguish between: (1) predictions that achieve low geometric error but correspond to chemically unstable conformations and (2) predictions that achieve both low geometric error and energetic stability.
> >
> > However, we believe assessing energy separately to demonstrate that REBIND does indeed predict such a stable conformation is important, and thus we present the energy assessment results. Kindly refer to our response in **Verification of Predicting “Low-Energy” Conformations** above.

---

> ### Comment · Reviewer_8Kvn · 2024-11-23
>
> Thanks for your response. I will raise my score.

---

> > ### Author Response · Authors · 2024-11-23
> >
> > We sincerely thank the reviewer for the reassessment! We are extremely glad that our responses have met and addressed your concerns.

---

### Official Review · Reviewer_P3Lq · 2024-11-03

**Soundness:** 3
**Presentation:** 3
**Contribution:** 1
**Rating:** 5
**Confidence:** 4

**Summary:**

The paper proposes a new attention mechanism for molecular conformation prediction tasks. This additional attention bias is constructed by extracting the estimated vdw force from the conformation predicted at an intermediate point in the network. The new architecture is then applied to the task of ground state prediction of molecules.

**Strengths:**

The addition of the force-based bias is well motivated. The usage of supervised predictions from the center of the model into the second half of the architecture seems somewhat novel. The paper is mostly clear (although certain parts could be summarized and simplified).

**Weaknesses:**

Although the paper does not have major flaws, the contribution seems somewhat incremental and the experimental validation unclear.

Regarding the contribution, while the authors present it as force-based graph rewiring, to me it appears that the authors simply add to the second half of the network a hand-crafted pairwise feature that is mapped with a linear layer to the attention bias of the transformer. Is my understanding correct? Are there key contributions I'm missing? With this in mind, while this is an interesting idea, I’m not sure it needs a 9 pages paper, especially if the experimental validation is not very strong.

Regarding the experimental validation, my main concern is the task definition and the way that it is evaluated. Molecular conformer generation methods do not aim to generate a single pose but multiple poses that represent the different low-energy conformation that the molecule can take. For this reason, the objective function with which they are typically trained are very different than the one used by the authors and the evaluation metrics are also significantly different and look not at the accuracy in capturing one pose but the whole set. A metric like the E-RMSE that the authors propose makes no sense to me: we are optimizing for the models to learn a weighted geometric average of the different conformations. For example, if we have two poses equally energetically favorable, this evaluation function would rank best the method that predicts the geometric average of these two poses, why would this be an interesting conformation? I believe the authors should adopt the metrics regularly used in the molecular conformer generation field that look at coverage & precision of the generated poses as well as their energetic properties.

**Questions:**

Are the RMSD/MAE/RMSE results corrected for molecular atom perturbations? If not, it is important to add such a correction.

---

> ### Author Response · Authors · 2024-11-21
> **Response to Reviewer P3Lq (1)**
>
> **W1. Regarding Novelty & Contribution**
>
> We thank the reviewer for the constructive feedback. However, we respectfully disagree with the statement that our contributions lie only in feature manipulation. We re-emphasize our contributions below.
>
> First, we make **novel observations that previous GNN-based conformer prediction approaches fall short on “low-degree atoms”**, showing large discrepancies in performance with respect to high-degree counterparts. This problem is crucial, as the number of low-degree atoms increases in larger and more complex molecular systems. We provide empirical results to this, where we have visualized the proportion of low-degree atoms ($\text{deg}_i^{\text{rel}}\in(0, 0.3]$) with respect to their molecular sizes in Figure 6 in Appendix. Combined with the observation of performance discrepancy between low-degree and high-degree atoms, this suggests that inaccuracies in predicting the positions of low-degree atoms are a major contributor to performance degradation in conformation prediction for large molecules. These observations emphasize the necessity of novel methods, such as REBIND, to address the challenges associated with low-degree atoms effectively.
>
> Second, our approach introduces a **force-based graph rewiring framework** that goes beyond simple feature addition. Unlike prior works that rely on ground-truth interatomic distances or 3D coordinates, we propose a self-guided mechanism using Lennard-Jones (LJ) potentials to dynamically identify and integrate critical non-bonded interactions. This rewiring is performed in a degree-compensating manner, mitigating the biases against low-degree atoms by selectively augmenting graph connectivity where it is most impactful. Our framework shares similarities in novelty with prior works on edge-rewiring to remedy oversquashing [1, 2, 3], edge-rewiring to enhance graph transformers [4, 5] and augmenting the graph itself for graph transformers [6], which modify input graphs to tackle specific tasks or issues.
>
> Finally, our force-based graph rewiring framework is **easily applicable to other backbones other than transformers as well** - demonstrating its versatility in bringing performance gains. On Table 4 in Appendix, we have shown that when incorporated with our framework, all baseline methods (GINE, GATv2, GraphGPS) show significant error reductions, up to 15% in RMSD.
>
> ---
> **References**
>
> [1] Locality-Aware Graph Rewiring in GNNs, 24’ ICLR
>
> [2] DRew: Dynamically Rewired Message Passing with Delay, 23’ ICML
>
> [3] FoSR: First-order spectral rewiring for addressing oversquashing in GNNs, 23’ ICLR
>
> [4] Global Self-Attention as a Replacement for Graph Convolution,  KDD '22
>
> [5] Path-Augmented Graph Transformer Network, 19’ ICML workshop
>
> [6] Gophormer: Ego-Graph Transformer for Node Classification, 21’ Arxiv preprint

---

> ### Author Response · Authors · 2024-11-21
> **Response to Reviewer P3Lq (2)**
>
> **W2. Regarding Task Definition & Evaluation**
>
> We understand the reviewer’s point. In response, we clarify that: (1) our work is about ground state conformer prediction, not about molecular conformer generation. (2) we already showcased all metrics used in prior work [1] that share our problem scenario, and (3) validation on adoption of E-RMSD.
>
> **(1) Clarification upon the problem scenario**
>
> The focus of REBIND is on predicting a single, most stable molecular conformation (the ground-state conformation) from a given 2D molecular graph. This problem setting is different from traditional conformer generation approaches, which aim to generate a diverse set of plausible low-energy conformations. Thus, unlike prior generation works of which was evaluated upon coverage or precision, our evaluation metric is the “prediction error with respect to the single lowest energy conformation”. Our problem definition is first proposed by [1], and its importance has been recently acknowledged in the research community. For instance, our ground-state conformer prediction task is recognized as an important problem in tasks such as crystal structure prediction [2-5], which aims to discover minimum-energy arrangement of constituent atoms in molecular crystals.
>
> **(2) Clarification upon utilized metrics**
>
> As mentioned, our task definition aligns with the benchmark proposed in GTMGC [1], where the goal is to predict the ground-state conformation characterized by the lowest energy. As such, evaluation metrics like D-MAE, D-RMSE, and C-RMSD (which we have included in our manuscript), which measure the accuracy of a single predicted conformation against the ground-truth conformation, are most suitable for our work.
>
> **(3) Validation of E-RMSD**
>
> As highlighted above, we consider a scenario where we specifically focus on predicting the single ground-state conformation. Given that our target conformation is the one exhibiting the lowest energy, it is crucial for models to not only minimize geometric errors (*e.g.*, RMSD) but also ensure that the predicted conformation is chemically plausible and energetically stable. The E-RMSD metric addresses this by penalizing models that predict low-error conformations but do not adhere to chemical stability principles. This weighting by energy helps distinguish between: (1) predictions that achieve low geometric error but correspond to chemically unstable conformations and (2) predictions that achieve both low geometric error and energetic stability. In summary, E-RMSD is a metric that incorporates energy-based factors to evaluate the energetic stability along with geometric errors, emphasizing our focus on the prediction of most stable conformation.
>
> ---
> **References**
>
> [1] GTMGC: Using Graph Transformer to Predict Molecule’s Ground-State Conformation, 24’ ICLR
>
> [2] Optimality guarantees for crystal structure prediction, 23’ Nature Computer Science
>
> [3] Crystal structure prediction of energetic materials and a twisted arene with Genarris and GAtor, 21’ CrystEngComm
>
> [4] Crystal structure prediction accelerated by Bayesian optimization, 18’ Physics Review Materials
>
> [5] Crystal Structure Prediction of Energetic Materials, 23’ Crystal Growth and Design

---

> ### Author Response · Authors · 2024-11-21
> **Response to Reviewer P3Lq (3)**
>
> **Q1. On Correction of Evaluation Results**
>
> We specify each metric’s correction process below.
> - **RMSD, E-RMSD Correction**: We ensure that both RMSD and E-RMSD are corrected for molecular atom perturbations. Specifically, we apply alignment (*e.g.*, correcting for rotations and translations) to the predicted and ground-truth conformations before calculating the RMSD, following the same practice in [1]. This ensures that the RMSD metric is invariant to such transformations.
> - **MAE and RMSE**: For MAE and RMSE, these metrics evaluate the errors in pairwise atomic distances, which are inherently perturbation-invariant. Therefore, following conventional practices and as done in prior works [1], we do not apply additional correction for these metrics.
>
> ---
> **References**
>
> [1] GTMGC: Using Graph Transformer to Predict Molecule’s Ground-State Conformation, 24’ ICLR

---

> ### Comment · Reviewer_P3Lq · 2024-11-23
> **Response to authors**
>
> Thank you for the response. A few comments:
> - I would recommend the authors to simplify the explanation of their method. I believe the method can be explained far more simply and the current explanation in the paper (and the response) is overly complicated.
> - I still find it hard to understand the goal of the E-RMSD metric and I would recommend the authors report the energy metrics separately from the RMSD metrics
> - From the description in the response the authors are correcting for translation and rotation invariance but not for permutation invariance, is this correct? Similarly, pairwise atomic distances are not inherently *permutation* invariant do the authors account for that?

---

> ### Author Response · Authors · 2024-11-23
>
> We thank the reviewer once more for the additional feedback in improving our manuscript. For the comments you've raised, we will address them as follows.
>
> **(1) Simplification of Method Section.**
>
> We will revise the method section of our manuscript, with focus on simplicity and conciseness.
>
> **(2) Concerns on E-RMSD.**
>
> We would like to clarify once more on the validity of our proposed metric, with experimental results for substantiation.
>
> The goal of REBIND is to predict the most stable molecular conformation (*i.e.* conformation *prediction task*) - where only a single conformation of **energy-wise global minima exists**. Note that this is in-line with protein structure prediction[1, 2, 3, 4]  and crystal structure prediction tasks [5, 6, 7]. This diverges from traditional conformation generation tasks, which aim to generate a *set* of conformations of similar energy levels, *i.e.* **set of conformations of energy-wise local minimas**.
> Since our task focuses on a single conformation, all our evaluation metrics focus on quantifying the errors between prediction (only a single prediction is made for a single input) and ground truth. E-RMSD incorporates “energy” into the quantification of geometrics errors, penalizing predictions that are energetically invalid. We emphasize once again, that since we consider a single output  and single ground truth conformation for each 2D molecular graph, and thus “optimizing the model to learn weighted geometric average of different conformations” does not happen.
>
> To (1) substantiate the assertion that our work focuses on predicting the most stable conformation and (2) to acknowledge the reviewer’s suggestion, we present the energy-based results below in Table 1. Specifically, we computed the average energy difference between predicted conformations and ground-state conformations across all baselines methods. The energy calculations were performed using the Merck Molecular Force Field(MMFF) in RDKit. It can be seen that REBIND consistently achieves to predict **most stable conformations** compared to baselines. We will add the following results in our revised manuscript.
>
> **(3) Concerns on Permutation.**
>
> We verify that our method along with GNN-based methods, maintain the ordering of atoms. Thus, it does not need corrections of ordering (permutation) during the computation of MAE and RMSE. We acknowledge the confusion, and will revise our manuscript to clarify this.
>
>
> Table 1. Energy difference of predicted conformations through different models from the ground-state.
> |$\text{KJ}\cdot\text{mol}^{-1}$|**QM9**|**Molecule3D (scaffold)**|**Molecule3D (random)**|**GEOM-DRUGS**|
> |---|---|---|---|---|
> |**GAT**|3.2842|91.2003|10.4040|271.8635|
> |**GINE**|9.2558|131.2986|5.5030|269.9960|
> |**GraphGPS**|0.3842|0.7318|1.3814|195.1740|
> |**GTMGC**|0.3882|0.5559|0.5557|1.7494|
> |**Ours**|**0.2632**|**0.3237**|**0.2736**|**1.3420**|
>
>
> ---
> **References**
>
> [1] Protein structure prediction with energy minimization and deep learning approaches, 23’ Natural Computing
>
> [2] Constructing effective energy functions for protein structure prediction through broadening attraction-basin and reverse Monte Carlo sampling, 19’ BMC Informatics
>
> [3] Rapid sampling of local minima in protein energy surface and effective reduction through a multi-objective filter, 13’ Proteome Science
>
> [4] Protein structure prediction by global optimization of a potential energy function, 99’ Proceedings of the National Academy of Sciences
>
> [5] Crystal Structure Prediction of Energetic Materials, 23’ Crystal Growth and Design
>
> [6] Minima Hopping Method for Predicting Complex Structures and Chemical Reaction Pathways, 18’ Handbook of Molecular Modeling
>
> [7] Crystal structure prediction accelerated by Bayesian optimization, 18’ Physics Review Materials

---

> > ### Author Response · Authors · 2024-11-25
> > **Gentle Reminder**
> >
> > Dear Reviewer P3Lq,
> >
> > We thank the reviewer once more for your constructive feedbacks.
> >
> > As the rebuttal period is nearing to its end, we kindly request you to review our responses upon your additional feedback, for we have faithfully addressed your concerns and suggestions.
> >
> > We once again appreciate your time and efforts in reviewing our paper.
> >
> > Sincerely,
> >
> > Authors.

---

> ### Comment · Reviewer_P3Lq · 2024-11-25
> **Response to authors**
>
> The authors' comment:
>
> *(3) Concerns on Permutation. We verify that our method along with GNN-based methods, maintain the ordering of atoms. Thus, it does not need corrections of ordering (permutation) during the computation of MAE and RMSE. We acknowledge the confusion, and will revise our manuscript to clarify this.*
>
> is makes the understanding of the method even more confusing to me. In particular: what is the output of the model? How is that translated into coordinated for the molecule conformation?
>
> Even if the output of some of the baselines is permutation equivariant (which I find weird but really depends on the questions asked above), the conformations of the molecule are not permutation equivariant, therefore the evaluation metric should account for this.

---

> ### Author Response · Authors · 2024-11-26
> **Follow-up Response to Reviewer P3Lq**
>
> We thank the reviewer for the additional feedback. To clarify, when we state that our method and the baseline methods “maintain the ordering of atoms,” this refers to the fact that the **atom ordering in the input graph is preserved throughout the forward pass**, and the predicted output aligns with this ordering. We provide additional clarification below.
>
> The input to the model includes:
>
> - **Adjacency matrix ($\mathbf{A} \in \mathbb{R}^{N \times N}$):** represents bond structure where element (i, j) is 1 when there exists a bond between node $i$ and $j$, and 0 otherwise.
> - **Node features ($\mathbf{X} \in \mathbb{R}^{N \times d}$):** represents features of a molecule with $N$ atoms where $i$ th row corresponds to node $i$.
>
> The output of the model is:
>
> - **Predicted Conformation matrix ($\hat{C} \in \mathbb{R}^{N \times 3}$):** where each row $\hat{C_i} \in \mathbb{R}^{3}$ represents the predicted 3D coordinates of node $i$, and $\hat{C_i}$ and $\mathbf{X}_i$ reference the same node $i$. Note that, the ground truth conformation $C$ is also already aligned with the ordering of $\mathbf{X}$, and consequently, $\hat{C_i}$ and $C_i$ corresponds to the same node $i$.
>
> From the predicted conformation $\hat{C}$ of a single molecule, we calculate MAE and RMSE with respect to the ground-truth interatomic distance $D_{ij} = ||C_i - C_j||_2$. The errors with respect to each node $i$ is defined as follows:
>
> $MAE(i)=\frac{1}{N-1}\sum_{ j\in \mathcal V\backslash \{i\} }  |\hat D_{ij} - D_{ij}|_1,$
>
> $RMSE(i) = \sqrt{\frac{1}{N-1}\sum_{j \in \mathcal V\backslash \{i\}} ( \hat D_{ij} - D_{ij} )^2},$
>
> where $\hat D_{ij} = ||\hat C_i - \hat C_j||_2$ is a predicted distance. During the whole process, the assignment of atom to node (*i.e.*, Nitrogen is assigned to node $i$) is the same between ground truth conformation $C$ and predicted conformation $\hat{C}$. Thus, it does not require “re-ordering” to match the ordering of atoms, or assignments.

---

> ### Comment · Reviewer_P3Lq · 2024-11-30
>
> I thank the authors for the additional reply but this still does not answer my original comment. The critical component that the authors seem to be missing from the replies is that molecules often have several symmetries. For example a simple molecule like benzene has a symmetry group with 12 elements (12 different permutations of the atoms resulting in the same molecule). This leads back to my question/comment:
>
> 1. Is the method permutation invariant? If not where is the permutation invariance broken?
> 2. The possible different permutations should be taken into account during evaluation to compute all the metrics.

---

> ### Author Response · Authors · 2024-12-01
> **Follow-up Response to Reviewer P3Lq (1)**
>
> We thank the reviewer for providing specific examples. We believe that there was a misunderstanding between us, and the example provided by the reviewer has alleviated this confusion.
>
> **Q1.** GNNs, including graph transformers with Laplacian encoding are inherently permutation equivariant [1, 2]. Our model, which is composed of transformer architecture with Laplacian encoding and adjacency matrix, is also permutation equivariant as well. The permutation invariance breaks when incorporating adjacency matrix as a bias term.
>
> **Q2-1. Metrics involving pairwise distances (MAE, RMSE)**
>
> We believe the reviewer’s question can be interpreted in the two following ways, regarding MAE and RMSE. 1) that the ordering of different atoms (i.e. Carbon, Nitrogen) that form the molecule will affect the results of MAE and RMSE. 2) that the symmetry of molecules (i.e. among the ordering that preserve the adjacency matrix of molecules, like the example of benzene with 12 symmetries) may affect the results of MAE and RMSE. We clarify upon both 1) and 2) below.
>
> **Scenario #1 - General permutation that changes the adjacency matrix**
>
> In our evaluation setting, permutations of atoms that change the adjacency matrix is not of consideration, for such ordering is fixed during evaluation. Consider the molecule H-C≡N-H. The ordering of the atoms can vary, but only a single input graph and atom ordering exists in our evaluation scheme, given by the RDKit’s `.GetAtoms()`. For example, the ordering is fixed to be C, N, H (connected to C), H (connected to N). Since all our baselines and REBIND preserves the ordering of geometrically different atoms, the output conformation aligns with the ground truth conformations, and thus does not require additional “alignment”. Thus, during evaluation, we only need to focus on handling molecular symmetries - alternations of atom orderings that do not alter the geometry nor the adjacency matrix - which is specified below.
>
> **Scenario #2 - molecular symmetry**
>
> In permutation scenarios that do not alter the adjacency matrix - *i.e.*, the symmetry in benzene, we provide detailed examples to demonstrate that, for RMSE and MAE, the results are the same.
>
> For simplicity, we consider ethane ($C2H6$).
>
> Ethane has two carbons ($C_1$ and $C_2$) connected by a single bond.
> - Each carbon is bonded to three hydrogens:
> - $C_1$ has $H_1$, $H_2$, $H_3$
> - $C_2$ has $H_4$, $H_5$, $H_6$
> - The order of atoms are: $C_1, C_2, H_1, H_2, H_3, H_4, H_5, H_6$.
>
> By symmetry, the hydrogens $H_1$ and $H_2$ around $C_1$, or $H_4$ and $H_6$ around $C_2$, can be swapped without changing the molecule’s geometry.
>
> Let’s say the original ground-truth pairwise distance matrix $D$ is as follows.
> $$
> D =
> \begin{bmatrix}
>   0 & 1.54 & 1.09 & 1.09 & 1.09 & 2.51 & 2.51 & 2.51 \cr
>   1.54 & 0 & 2.51 & 2.51 & 2.51 & 1.09 & 1.09 & 1.09 \cr
>   \vdots & \vdots & \vdots & \vdots & \vdots & \vdots & \vdots & \vdots
> \end{bmatrix}
> $$
> Then given the molecular graph, the predicted pairwise distance matrix $\hat D$ can be as follows.
> $$
> \hat D =
> \begin{bmatrix}
>   0 & 1.55 & 1.08 & 1.10 & 1.07 & 2.50 & 2.53 & 2.49 \cr
>   1.55 & 0 & 2.50 & 2.52 & 2.49 & 1.11 & 1.08 & 1.10 \cr
>   \vdots & \vdots & \vdots & \vdots & \vdots & \vdots & \vdots & \vdots
> \end{bmatrix}
> $$
> Consider a scenario where the hydrogen mentioned above are swapped; $H_1$ ↔ $H_2$, $H_4$ ↔ $H_6$, in the prediction matrix. The swapping results in the matrix $\hat D_{swapped}$ is as below.
> $$
> \hat D_{swapped} =
> \begin{bmatrix}
>   0 & 1.55 & \mathbf{1.10} &  \mathbf{1.08} & 1.07 &  \mathbf{2.49} & 2.53 &  \mathbf{2.50} \cr
>   1.55 & 0 &  \mathbf{2.52} &  \mathbf{2.50} & 2.49 &  \mathbf{1.10} & 1.08 &  \mathbf{1.11} \cr
>   \vdots & \vdots & \vdots & \vdots & \vdots & \vdots & \vdots & \vdots
> \end{bmatrix}
> $$
> Now, when we compute the MAE for both cases, we can see that the MAE is the same for both cases of $MAE(\hat D - D)$ and $MAE(\hat D_{swapped} - D)$, since the errors are averaged in whole (see the definition of MAE in our previous response). Thus, as demonstrated from the exemplar, we can see that for such symmetries that do not alter the adjacency matrix of the molecular graph, the MAE remains the same. This holds for RMSE as well, which is computed in a similar manner.

---

> > ### Author Response · Authors · 2024-12-01
> > **Follow-up Response to Reviewer P3Lq (2)**
> >
> > **Q2-2. Metric Involving RMSD**
> >
> > We acknowledge our initial response regarding RMSD calculations is not profoundly stated. While our evaluation process accounts for rotation and translation during alignment, it also takes into account **symmetrical permutations**, as noted by the reviewer, and returns the optimal error among the possible permutations. Specifically, we utilized RDKit’s `rdkit.Chem.rdMolAlign.GetBestRMS` function to compute RMSD. According to RDKit’s documentation:
> >
> > “Returns the optimal RMS for aligning two molecules, **taking symmetry into account**. This function will attempt to **align all permutations** of matching atom orders in both molecules.” [3]
> >
> > Thus, the symmetrical permutations noted by the reviewer are indeed accounted for in our evaluation process. We will update our manuscript accordingly to address this.
> >
> > ---
> >
> > **References**
> >
> > [1] Approximately Equivariant Graph Networks, 23’ NeurIPS
> >
> > [2] Equivariant and Stable Positional Encoding for More Powerful Graph Neural Networks, 22’ ICLR
> >
> > [3] https://www.rdkit.org/docs/source/rdkit.Chem.rdMolAlign.html

---

> ### Author Response · Authors · 2024-12-03
> **Dear Reviewer P3Lq,**
>
> As the discussion period is coming to a close, we await your response.
>
> We have carefully addressed your concerns, and hope our responses have resolved any issues or questions.
>
> If you have any additional comments or concerns, please don't hesitate to let us know.
>
> Sincerely,
>
> Authors

---

### Official Review · Reviewer_5PhF · 2024-11-05

**Soundness:** 3
**Presentation:** 3
**Contribution:** 3
**Rating:** 6
**Confidence:** 3

**Summary:**

This work introduces REBIND for ground-state molecular conformation prediction using force-based graph rewiring. By adding edges based on Lennard-Jones potential, REBIND captures non-bonded interactions between low-degree atoms. This framework includes an encoder-decoder graph transformer to first estimate atomic forces and then integrate those interactions into refined molecular graphs. Experimental results show substantial performance improvements, demonstrating REBIND’s effectiveness and scalability across multiple benchmarks and molecular sizes.

**Strengths:**

This work presents a novel contribution to molecular conformation prediction, primarily through its force-based graph rewiring mechanism. By incorporating the Lennard-Jones potential for non-bonded interactions, REBIND effectively addresses the low-degree atoms, which often suffer from high prediction errors in conventional models. The proposed force-aware edges between atoms provide an extra solution for creating molecular graphs besides relying on bonds and pairwise distances. REBIND could inspire further research in both conformer generation and broader applications in molecular property prediction.

This work contains rigorous experimental design and comprehensive evaluation on well-established datasets (QM9, Molecule3D, GEOM-DRUGS). The quantitative results are compelling, showing substantial improvements over baseline methods across multiple metrics.

In terms of clarity, the paper is well-structured, with clear explanations of the related concepts.

**Weaknesses:**

The authors didn't provide an efficiency evaluation of the proposed method. It would be better to know about the computational burden by introducing the force-aware adjacency matrices when compared to the ablated settings in ablation study.

When introducing the prior works, the authors didn't fully cover the previous GNN works [1-3] in this field that make efforts to model non-bonded interactions. Unlike GTMGC, those works already use non-linear functions of distance with atom type-specific coefficients when modeling non-bonded interactions. It is better to mention them to strengthen the coverage of prior related works.

[1] Kosmala, Arthur, et al. "Ewald-based long-range message passing for molecular graphs." International Conference on Machine Learning. PMLR, 2023.

[2] Zhang, Shuo, Yang Liu, and Lei Xie. "A universal framework for accurate and efficient geometric deep learning of molecular systems." Scientific Reports 13.1 (2023): 19171.

[3] Li, Yunyang, et al. "Long-Short-Range Message-Passing: A Physics-Informed Framework to Capture Non-Local Interaction for Scalable Molecular Dynamics Simulation." International Conference on Learning Representations. 2024.

**Questions:**

Are the force-aware adjacency matrices only computed once for decoding? If so, have the authors tried recycling the predicted molecular conformation C several times, i.e. using the final predicted C to compute new force-aware adjacency matrices and doing the decoding again? The new force-aware adjacency matrices could capture more accurate atomic pairs, so as to enable even better conformation predictions.

---

> ### Author Response · Authors · 2024-11-21
> **Response to Reviewer 5PhF (1)**
>
> **W1. Computational Burden Assessment**
>
> We thank the reviewer for this valuable observation. To address this, we report the wall-clock computation time during the forward and backward pass of the training process (including gradient computation) in our implementation. We compare the time complexity between 1. incorporating $A^{bond}$  alone, 2. $A^{bond}$ + $D^{row-sub}$ (*i.e.*, GTMGC [1]), 3. $A^{bond}$ + $A_{near}$ (*i.e.*, adjacency matrix constructed via close distance), and 4. REBIND. Note that, the results are averaged upon 50 gradient update steps for each dataset, for they have varying molecular sizes (8, 15, and 25 average molecular size for each dataset).
>
> The results are summarized in Table 1 below. On average, our method only sacrifices an additional 0.03 seconds of computational overhead for computing the Lennard-Jones (LJ) potential matrix, compared to the GTMGC.
>
> Table 1. Wall-clock time analysis. The comparison is conducted on the machine with NVIDIA GeForce RTX 3090 GPU/Intel(R) Xeon(R) Gold 5215 CPU @2.50GHz.
> |           Sec.             | QM9     |      |       | Molecule3D  |     |    | Geom-Drugs  |     |    |
> |------|------|---------|---------|-------|-------|--------|--------|---------|------|
> |                        | forward | backward | total   | forward | backward | total   | forward | backward | total   |
> | $A^{bond}$                | 0.0287  | 0.0379   | 0.0666  | 0.0298  | 0.0367   | 0.0665  | 0.0943  | 0.105    | 0.1993  |
> | **GTMGC**        | 0.0311  | 0.039    | 0.0701  | 0.0307  | 0.0467   | 0.0774  | 0.12888 | 0.1442   | 0.27308 |
> | $A^{bond}$ + $A _{near}$        | 0.0594  | 0.0403   | 0.0997  | 0.0609  | 0.0431   | 0.104   | 0.1691  | 0.092    | 0.2611  |
> | **Ours**                  | 0.0637  | 0.0428   | 0.1065  | 0.0691  | 0.0412   | 0.1103  | 0.1947  | 0.109    | 0.3037  |
>
> ---
> **References**
>
> [1] GTMGC: Using Graph Transformer to Predict Molecule’s Ground-State Conformation, 24’ ICLR

---

> > ### Author Response · Authors · 2024-11-21
> > **Response to Reviewer 5PhF (2)**
> >
> > **W2. Discussion with Prior Works**
> >
> > We thank the reviewer for helping us in improving our manuscript. Below we highlight the key differences with each referred paper. We will add the following discussion and comparison with prior works to our revised manuscript.
> >
> > **1. Ewald-based long-range message passing for molecular graphs [1]**
> >
> > This paper proposes Ewald Message Passing, a framework that integrates the Ewald summation technique into GNNs to model long-range interactions like electrostatic and van der Waals forces in molecular systems, mainly in periodic systems. However, Ewald MP assumes obtaining precise 3D geometric information of ground truth 3D coordinates to perform Fourier decomposition and calculation of long-range electrostatic interactions, thus diverging from our problem scenario where we do not have access to such ground truth coordinates. Furthermore, unlike Ewald MP which performs a comprehensive summation of potential across all atomic pairs in the lattice space using Fourier transformation, REBIND focuses on the most relevant atomic pairs based on LJ potential, selectively modeling critical attractive and repulsive forces at its current prediction of the model, enabling self-revision of its predictions.
> >
> > **2. A universal framework for accurate and efficient geometric deep learning of molecular systems [2]**
> >
> > PAMNet models molecular systems using a physics-inspired multiplex graph framework that separates local and non-local interactions, relying on accurate 3D coordinates, pairwise distances, and angles for its computations. As in Ewald MP [1], this diverges from our problem scenario of REBIND where it does not have any ground truth conformations to utilize; and instead uses a self-guided framework. Specifically, one cannot construct the $G_{global}$ without the access to ground truth 3D conformations - where it extensively utilizes information such as pairwise distances, and bonding angles.
> >
> > **3.  Long-Short-Range Message-Passing: A Physics-Informed Framework to Capture Non-Local Interaction for Scalable Molecular Dynamics Simulation [3]**
> >
> > The Long-Short-Range Message-Passing (LSR-MP) framework is a physics-informed GNN architecture designed for molecular dynamics simulation, where they incorporate both short-range radius-based message passing modules with a long-range bipartite message passing module through fragment representations. Like former works above [1, 2], this requires the extensive information of ground-truth conformation in order to generate $g_{short}$ and $g_{long}$, which is constructed by thresholding the interatomic distances. REBIND on the other hand, does not require specific thresholding of distances; it adds interactions between atoms that most affect the conformation
> > up to the maximum possible degree, based on the interatomic forces. Note that this is more relevant in conformation prediction where the play of interatomic forces judge the ground-state conformation of a molecule. Additionally, our method does not require additional hyperparameters for thresholding.
> >
> > ---
> > **References**
> >
> > [1] Ewald-based long-range message passing for molecular graphs, 23’ ICML
> >
> > [2] A universal framework for accurate and efficient geometric deep learning of molecular systems, 23’ Scientific Reports
> >
> > [3] Long-Short-Range Message-Passing: A Physics-Informed Framework to Capture Non-Local Interaction for Scalable Molecular Dynamics Simulation, 24’ ICLR

---

> > > ### Author Response · Authors · 2024-11-21
> > > **Response to Reviewer 5PhF (3)**
> > >
> > > **Q1. Recycling of $A_{force}$**
> > >
> > > We appreciate the reviewer’s insightful suggestion. During the inference phase of the pretrained REBIND network, it is indeed possible to recycle the outputs from the decoder to recompute the LJ potential matrix, and forward it through the decoder once again. This process can be repeated N times, as suggested by the reviewer. We have ablated the number of iterations (N) and present the results below.
> > > With iterative recycling, we observe performance improvements in RMSD and E-RMSD. However, we see that the performance typically saturates after the second iteration. Specifically, on the QM9 and GEOM-Drugs dataset, the performance gain is non-monotonic of the iteration number.
> > >
> > > Table 2. Performance of REBIND with iterative recycling, up to 5 times.
> > > | Dataset                | QM9                       |                             | Molecule3D                |                             | GEOM-DRUGS                |                             |
> > > |------------------------|---------------------------|-----------------------------|---------------------------|-----------------------------|---------------------------|-----------------------------|
> > > | **Metric**             | **RMSD**                  | **E-RMSD**                  | **RMSD**                  | **E-RMSD**                  | **RMSD**                  | **E-RMSD**                  |
> > > | 1 (Default)            | 0.3210                    | 0.6010                      | 0.6670                    | 1.1820                      | 1.3960                    | 2.6020                      |
> > > | 2                      | 0.3028                    | **0.6091**                  | 0.6484                    | 1.1827                      | **1.3230**                    | **2.4320**                      |
> > > | 3                      | 0.3026                    | **0.6091**                  | 0.6445                    | 1.1756                      | 1.3250                    | 2.4390                      |
> > > | 4                      | 0.3026                    | **0.6091**                  | 0.6427                    | 1.1728                      | 1.3260                    | 2.4390                      |
> > > | 5                      | **0.3025**                    | **0.6091**                  | **0.6420**                    | **1.1714**                      | 1.3270                    | 2.4420                      |

---

> > > > ### Author Response · Authors · 2024-11-25
> > > > **Gentle Reminder**
> > > >
> > > > Dear Reviewer 5PhF,
> > > >
> > > > We thank the reviewer once more for your constructive feedbacks.
> > > >
> > > > As the rebuttal period is nearing to its end, we kindly request you to review our responses.
> > > >
> > > > We appreciate your time and efforts in reviewing our paper.
> > > >
> > > > Sincerely,
> > > >
> > > > Authors.

---

> > > > > ### Comment · Reviewer_5PhF · 2024-11-26
> > > > >
> > > > > Thank you for the response and the additional results provided! My questions and concerns including computational burden, discussion with prior works, and recycling strategy have been all addressed accordingly.

---

> > > > > > ### Author Response · Authors · 2024-11-27
> > > > > >
> > > > > > We sincerely thank the reviewer for the positive opinion for our work! We are glad that our responses have met and addressed your concerns.

---

### Author Response · Authors · 2024-11-21
**Global Response**

Dear Reviewers and AC,

We sincerely appreciate your dedicated time and insightful feedback in improving our manuscript.

As acknowledged by the reviewers, we introduce a new observation that identifies the limitation of prior works on addressing the inaccuracies present in low-degree atoms (5PhF, P3Lq, 8Kvn, twMi). To tackle this challenge, we propose REBIND, a novel (5PhF, 8Kvn, twMi) framework that utilizes Lennard-Jones Potential to rewire edges in a degree-compensating manner. Extensive evaluations demonstrate our method’s efficacy (5PhF, 8Kvn, twMi) and versatility to diverse benchmarks and GNN architectures (5PhF, 8Kvn, twMi).

In accordance with your constructive comments, we have faithfully executed the following additional experiments and discussions:
- **Computational burden assessment** (forward, backward, and total computation time): our method only sacrifices an additional 0.03 sec. of total computation time, compared to baseline.
- **Recycling of $A_{force}$**  (*i.e.*, using final predicted C to recompute $A_{force}$ to pass into the decoder) to yield further improvements.
- **Clarification upon our problem scenario**; our focus being on predicting the single conformation of molecules at the global energy minima (*i.e.*, ground state conformer prediction).
- Empirical verification that addressing errors in low-degree atoms is crucial for larger molecules (Kindly refer to Figure 6 in Appendix of our revised manuscript).
- Energy-wise verification that REBIND indeed produces ground-state conformations.
- **Extensive ablation** upon the efficacy of $A_{force}$ through comparisons with combinations of $A^{bond}$, $D^{row-sub}$, and $A_{near}$.

For changes made in our manuscript, we have marked them as *blue* for your convenience.

Thank you for your guidance and support.

Sincerely,

Authors.

---

### Meta-Review · Area_Chair_xb6o · 2024-12-22

**Metareview:**

The paper studies an interesting and important problem and received positive support from majority of the reviewers.

**Additional Comments On Reviewer Discussion:**

Most of the issues have been resolved during rebuttals. One of the reviewers raised questions on symmetry of the methods, and the authors have replied extensively.

---

### Decision · Program_Chairs · 2025-01-22

Accept (Poster)